# The underlying structures of self-attention: symmetry, directionality, and emergent dynamics in Transformer training

Matteo Saponati [1] [*]  Pascal Sager [2] [1] [*]  Pau Vilimelis Aceituno [1]  Thilo Stadelmann [2] [3]  Benjamin Grewe [1] [4]

## Abstract

Self-attention is essential to Transformer architectures, yet how information is embedded in the self-attention matrices and how different objective functions impact this process remains unclear. We present a mathematical framework to analyze self-attention matrices by deriving the structures governing their weight updates. Using this framework, we demonstrate that bidirectional training induces symmetry in the weight matrices, while autoregressive training results in directionality and column dominance. Our theoretical findings are validated across multiple Transformer models — including ModernBERT, GPT, LLaMA3, and Mistral — and input modalities like text, vision, and audio. Finally, we apply these insights by showing that symmetric initialization improves the performance of encoder-only models on language tasks. This mathematical analysis offers a novel theoretical perspective on how information is embedded through self-attention, thereby improving the interpretability of Transformer models.

## 1. Introduction

Transformer models now achieve state-of-the-art performance across a wide range of tasks and domains (Radford et al., 2019; Dosovitskiy et al., 2021; Radford et al., 2023). Despite their success, the internal mechanisms governing their decision-making processes remain poorly understood, raising concerns regarding model alignment, reliability, and safety (Wang et al., 2023; Yao et al., 2024). A key challenge

in understanding these models is unraveling the structures of self-attention, which is essential to Transformer architectures. Current literature largely overlooks the nature of the self-attention weight matrices during autoregressive training, where the model predicts the next token in a sequence given previous ones (Radford et al., 2019; Black et al., 2021; Touvron et al., 2023) and bidirectional training, where the model predicts a missing token given the full sequence (Devlin et al., 2019; Bao et al., 2022; Warner et al., 2024). Understanding self-attention requires answering two fundamental questions: How can we interpret the structures learned in the self-attention matrices? What is the impact of different objective functions on these matrices?

Previous work used sparse auto-encoders to identify interpretable features (Huben et al., 2024; Bricken et al., 2023), circuit analysis to interpret Transformer components (Olah et al., 2020; Elhage et al., 2021; Olah, 2022), and techniques like the logit lens to analyze self-attention mechanisms (Geva et al., 2021; Dar et al., 2023) (for a detailed discussion, see Section 5). However, these methods do not reveal the structural patterns in self-attention matrices or the transformations they encode. Crucially, how autoregressive and bidirectional training shape specific weight structures remains unclear.

To address this gap, we introduce a novel framework for analyzing self-attention matrices and understanding how different objective functions define their weight updates. We then use this framework to derive understandable mathematical structures that should emerge from such updates. Finally, we verify these interpretable structures numerically on many pre-trained and custom models across different modalities, supporting the universality of our results. Identifying these universal structures is fundamental not only for improving the performance of Transformer models, but also for their safety, alignment, and interpretability (Olah, 2022).

Specifically, we connect the matrix $\mathbf{W}_{qk} = \mathbf{W}_q \mathbf{W}_k^\top$ of self-attention with bilinear forms, offering novel insights compared to studying query and key matrices alone. We reveal structured patterns in the implicit weight updates of $\mathbf{W}_{qk}$, uncovering key differences between encoder-only and decoder-only models:

---

[*]Equal contribution  [1]Institute of Neuroinformatics, ETH Zürich and University of Zürich, Zürich, Switzerland [2]Centre for Artificial Intelligence, Zürich University of Applied Sciences, Winterthur, Switzerland [3]ECLT European Centre for Living Technology, Venice, Italy [4]ETH AI Center, Zürich, Switzerland. Correspondence to: Matteo Saponati <masapo@ini.ethz.ch>, Benjamin Grewe <bgrewe@ini.ethz.ch>.

*Proceedings of the 42ⁿᵈ International Conference on Machine Learning*, Vancouver, Canada. PMLR 267, 2025. Copyright 2025 by the author(s).

1. *Decoder-only models*: Training with an autoregressive objective produces a few columns with disproportionately high norms, introducing directionality in $\mathbf{W}_{qk}$.
2. *Encoder-only models*: Bidirectional optimization induces symmetric structures in $\mathbf{W}_{qk}$, reflecting the balanced nature of the training objective.
3. We validate these theoretical findings across diverse Transformer architectures and input modalities, showing that they generalize across models and tasks.
4. Empirically, we find that symmetric structures in $\mathbf{W}_{qk}$ enhance training efficiency for encoder-only models, leading to higher accuracy and faster convergence in language tasks.

## 2. Autoregressive and bidirectional training leads to directional and symmetric weight updates

In this section, we introduce a novel framework that links the self-attention matrices $\mathbf{W}_{qk}$ to bilinear forms, enabling us to analyze how an objective function influences their structure. This approach reveals fundamental patterns in $\mathbf{W}_{qk}$ that are not apparent when examining $\mathbf{W}_q$ and $\mathbf{W}_k$ separately. For example, we prove that autoregressive and bidirectional training induce directional and symmetric structures in the $\mathbf{W}_{qk}$ matrices. In the following sections, we define and formalize these concepts.

### 2.1. Interpreting self-attention with bilinear forms

Self-attention (Vaswani et al., 2017; Radford et al., 2019) is a type of score function $A : \mathbf{R}^{N,d} \times \mathbf{R}^{N,d} \to \mathbf{R}^{N,N}$ that maps a sequence of $N$ token embeddings with dimension $d$ into a matrix of attention scores. Except for the row-wise softmax function $\sigma(\cdot)$, self-attention is a linear transformation of the embedded tokens. In particular,

$$
\begin{aligned}
\mathbf{A}(\mathbf{X}) &= \sigma\left(\frac{1}{\sqrt{d}}\,\hat{\mathbf{A}}(\mathbf{X})\right) \\
&= \sigma\left(\frac{1}{\sqrt{d}}\,\mathbf{Q}\mathbf{K}^T\right) \\
&= \sigma\left(\frac{1}{\sqrt{d}}\,\mathbf{X}\mathbf{W}_{qk}\mathbf{X}^T\right),
\end{aligned}
\tag{1}
$$

where $\hat{\mathbf{A}}(\mathbf{X})$ is the linear part of self-attention (raw unscaled attention scores), $\mathbf{X} = [\mathbf{x}_1^\top, \ldots, \mathbf{x}_N^\top] \in \mathbb{R}^{N,d}$ is the sequence of $N$ token embeddings $\mathbf{x}_i \in \mathbb{R}^d$, and $\mathbf{W}_{qk} = \mathbf{W}_q \mathbf{W}_k^\top \in \mathbb{R}^{d,d}$. This equation shows that the linear transformation $\mathbf{W}_q$ and $\mathbf{W}_k$ are always combined to compute attention scores with one single matrix $\mathbf{W}_{qk}$. While the matrices $\mathbf{W}_q$ and $\mathbf{W}_k$ are defined separately for computational efficiency, this formulation remains mathematically equivalent (see also Elhage et al., 2021; Olsson et al., 2022; Dar et al., 2023).

We observe that $\mathbf{X}\mathbf{W}_{qk}\mathbf{X}^\top$ corresponds to a bilinear form (see Definition A.2). Specifically, the entry $\hat{\alpha}_{ij} = [\hat{\mathbf{A}}]_{ij}$ can be formulated in two equivalent ways: (1) as the canonical dot product between a query $\mathbf{q}_i$ and a key $\mathbf{k}_j$ (like in standard Transformer models), or (2) as the dot product between tokens $\mathbf{x}_i$ and $\mathbf{x}_j$ under the bilinear form $\mathbf{W}_{qk}$,

$$
\hat{\alpha}_{ij} = \langle \mathbf{q}_i,\, \mathbf{k}_j \rangle = \langle \mathbf{x}_i, \mathbf{W}_{qk}\mathbf{x}_j \rangle = \langle \mathbf{x}_i, \mathbf{x}_j \rangle_{\mathbf{W}_{qk}}.
\tag{2}
$$

Intuitively, this equivalence shows that the matrices $\mathbf{W}_q$ and $\mathbf{W}_k$ together define an alternative metric in the embedding space, which quantifies the score of $\mathbf{x}_i$ and $\mathbf{x}_j$ without explicitly constructing the query and key vectors. Let us consider a specific input token $\mathbf{x}_i \in \mathbb{R}^d$. The self-attention layer maps it to an updated token $\hat{\mathbf{x}}_i \in \mathbb{R}^d$ as follows,

$$
\hat{\mathbf{x}}_i^\top = \mathbf{x}_i^\top + \sum_{j=1}^{N} \hat{\alpha}_{ij}\,\mathbf{x}_j \mathbf{W}_v,
\tag{3}
$$

where the coefficients $\{\hat{\alpha}_{ij}\}$ in the second term represent the projection of $\mathbf{x}_i$ onto span$\{\mathbf{X}\}$ (the subspace spanned by $\mathbf{X} = \{\mathbf{x}_0^\top, \mathbf{x}_1^\top, \ldots, \mathbf{x}_N^\top\}$) in the transformed embedding space defined by $\mathbf{W}_{qk}$,

$$
\sum_j \hat{\alpha}_{ij}\,\mathbf{x}_j = \sum_j \langle \mathbf{x}_i, \mathbf{x}_j \rangle_{\mathbf{W}_{qk}}\,\mathbf{x}_j.
\tag{4}
$$

Because the softmax function preserves the order of its input values, the ranking of the raw attention scores $\hat{\alpha}_{ij}$ is maintained in the final normalized scores $\alpha_{ij}$,

$$
\alpha_{ij} < \alpha_{ij'} \Leftrightarrow \hat{\alpha}_{ij} < \hat{\alpha}_{ij'} \quad \forall i, j, j'.
\tag{5}
$$

As a result, the attention weights $\alpha_{ij}$ define a convex combination of the token vectors $\mathbf{x}_j$, meaning the output lies within the convex hull of the input sequence Conv$(\mathbf{X}) \subset$ span$\{\mathbf{X}\}$ (see also Appendix A.2). We note that $\mathbf{W}_v$ is a linear transformation that is applied independently to each projection in the sum and thus does not influence our derivation.

In the following sections, we demonstrate how this equivalent formulation of self-attention provides a useful framework for analyzing the training of Transformer models. Specifically, we show that the choice of the objective function such as autoregressive prediction (Radford et al., 2019) or bidirectional training (Devlin et al., 2019) produces distinct structural patterns in $\mathbf{W}_{qk}$.

### 2.2. Deriving the gradients of self-attention with bilinear forms

To show the connection between the objective function and the structures of self-attention matrices, we derive a convenient formulation for the weight update of $\mathbf{W}_{qk}$. We first formulate a sequence modeling problem with self-supervised

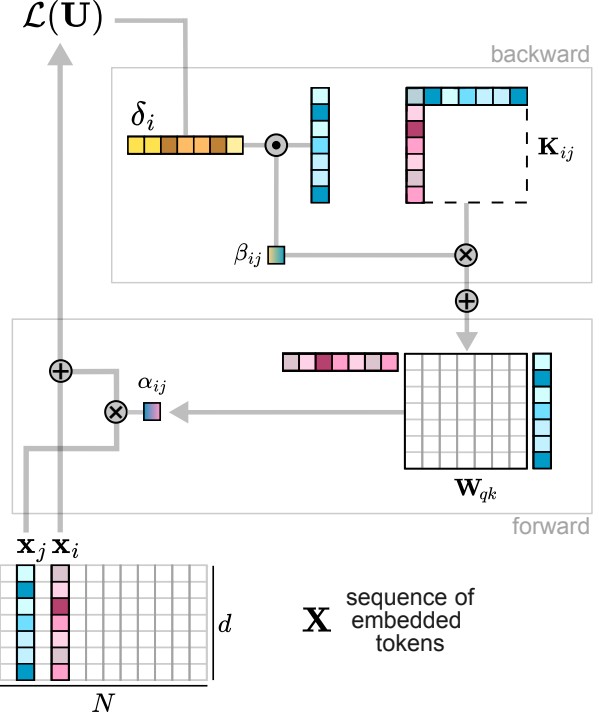

$\mathcal{L}(\mathbf{U})$

*Figure 1.* Illustration of the computation of the self-attention score between token $\mathbf{x}_i$ and token $\mathbf{x}_j$ (forward pass, see Equation (3)), and its corresponding contribution to the weight update of $\mathbf{W}_{qk}$ (backward pass, see Equation (8) and Equation (9)). The symbols "$\oplus$", "$\otimes$", and "$\odot$" refer to the addition, multiplication, and dot product operations, respectively.

training as follows. Let $U = \{t_1, \ldots, t_N\}$ be a sequence of $N$ tokens. For each position $i$, let $\mathcal{D}_i \subseteq \{1, 2, \ldots, N\}$ denote the *conditioning set*, that is, the set of indices corresponding to the tokens used to predict the token $t_i$. This formulation makes it possible to isolate and analyze the contribution of each token $t_j \in \mathcal{D}_i$ to the prediction of $t_i$. Let $\mathcal{L}(U)$ be the negative log-likelihood of each token $t_i$, expressed as,

$$\mathcal{L}(U; \mathcal{W}) = \sum_{i=1}^{N} \mathcal{L}(t_i) = -\sum_{i=1}^{N} \log p_{\mathcal{W}}(t_i \mid t_{\mathcal{D}_i}), \quad (6)$$

where $\mathcal{W}$ is the set of trainable parameters, and $p_{\mathcal{W}}(t_i \mid t_{\mathcal{D}_i})$ are the conditional densities parameterized by the model (see also Appendix A.3).

Here, we demonstrate that the updates to the matrix $\mathbf{W}_{qk}$ follow a structured pattern: the contribution of the token $t_j$ in the prediction of the embedding $t_i$ results in adding a rank-1 matrix $\mathbf{K}_{ij}$ to the matrix $\mathbf{W}_{qk}$. As a result, the total weight update to $\mathbf{W}_{qk}$ is expressed as a linear combination of these rank-1 matrices. We formalize this observation in the following proposition.

**Proposition 2.1.** *(**The implicit weight update as sum of rank-1 matrices**). Let $U = \{t_1, \ldots, t_N\}$ be a sequence of $N$ tokens, and let $\mathcal{L}(U; \mathcal{W})$ denote the negative log-likelihood of the sequence under a Transformer model with learnable parameters $\mathcal{W}$,*

$$\mathcal{L}(U; \mathcal{W}) = \sum_i \mathcal{L}(t_i) = -\sum_i \log p_{\mathcal{W}}(t_i \mid t_{\mathcal{D}_i}), \quad (7)$$

*where $\mathcal{D}_i \subseteq \{1, \ldots, N\}$ is the set of indices used to condition the prediction of token $t_i$. Let the self-attention function be defined as in Equation (1). Following the gradient of $\mathcal{L}(U; \mathcal{W})$, the implicit gradient-based update to the bilinear weight matrix $\mathbf{W}_{qk}^l$ at the $l$-th self-attention layer is proportionally equivalent to:*

1. *The sum of contributions from each token $t_j : j \in \mathcal{C}_i$ used to predict $t_i$,*

$$\Delta \mathbf{W}_{qk}^l \propto \sum_i \sum_{j \in \mathcal{C}_i} \Delta \mathbf{W}_{qk}^l \Big|_{t_i \leftarrow t_j} = \sum_i \sum_{j \in \mathcal{C}_i} \beta_{ij}^l \mathbf{K}_{ij}^{l-1}, \quad (8)$$

*where $\mathcal{C}_i$ denotes the set of context indices for predicting $t_i$.*

$$\Delta \mathbf{W}_{qk} \propto \sum_{i \in \mathcal{P}_j} \sum_j \beta_{ij} \mathbf{K}_{ij}$$

2. *The sum of contributions from each prediction target $t_i : i \in \mathcal{P}_j$ when predicted with $t_j$,*

$$\Delta \mathbf{W}_{qk}^l \propto \sum_{i \in \mathcal{P}_j} \sum_j \Delta \mathbf{W}_{qk}^l \Big|_{t_i \leftarrow t_j} = \sum_{i \in \mathcal{P}_j} \sum_j \beta_{ij}^l \mathbf{K}_{ij}^{l-1}, \quad (9)$$

*where $\mathcal{P}_j$ denotes the set of tokens for which $t_j$ is included in the context.*

*Here, $\beta_{ij}^l$ is a scalar that quantifies the contribution of the token embedding $\mathbf{x}_j$ to the prediction error for $\mathbf{x}_i$ at the $l$-th layer, and the matrix $\mathbf{K}_{ij}^{l-1} \in \mathbb{M}_d$ is a rank-1 matrix defined by the outer product of the token embeddings at the previous layer,*

$$\mathbf{K}_{ij}^{l-1} = \mathbf{x}_i^{l-1} \mathbf{x}_j^{l-1 \top}. \quad (10)$$

We provide proof for this proposition with related remarks in Appendix A.4, and an illustrative description of the forward and backward pass in Figure 1.

### 2.3. How context and prediction impact the gradient differently

Next, we show how the formulation of $\Delta \mathbf{W}_{qk}$ enables the analysis of the contribution of any given token $t^*$ to the weight updates and how this affects the properties of $\mathbf{W}_{qk}$. Indeed, Proposition 2.1 indicates that a token $t^*$ impacts the updates of $\mathbf{W}_{qk}$ differently when serving as context for predicting other tokens or being itself predicted.

When a token $t^*$ serves as context ($t_j = t^*$), the embeddings of all predicted tokens contribute to the column space of $\mathbf{W}_{qk}$, where the update of every $k$-th column $\mathbf{w}_{\cdot,k}$ is given by

$$\Delta\mathbf{w}_{\cdot,k}\Big|_{t_j=t^*} = [\mathbf{x}_{t^*}]_k \left( \sum_{i\in\mathcal{P}_t} \beta_{i*}\mathbf{x}_i \right). \tag{11}$$

Only the embedding of $t^*$ is instead added to the row space, where the update of every $m$-th row $\Delta\mathbf{w}_{m,\cdot}$ is given by

$$\Delta\mathbf{w}_{m,\cdot}\Big|_{t_i=t^*} = \left( \sum_{i\in\mathcal{P}_t} \beta_{i*}[\mathbf{x}_i]_k \right) \mathbf{x}_{t^*}. \tag{12}$$

Intuitively, using $t^*$ as context increases the dimensionality of the column space proportionally to the embeddings of the predicted tokens, while reducing the row space along the direction of the embedding of $t^*$. Conversely, when $t^*$ is being predicted ($t_i = t^*$), all token embeddings from the context are added to the row space of $\mathbf{W}_{qk}$, while only the embedding of $t^*$ is added to the column space. Consequently, the role a token plays during training affects its contribution to the weight update differently. We formalize this in the following proposition.

**Proposition 2.2.** *(Different implicit updates for context and prediction). Let $U = \{t_1, \ldots, t_N\}$ be a sequence of tokens and let $\mathbf{x}_i \in \mathbb{R}^d$ be the embedding of the $i$-th token. Let $\Delta\mathbf{W}_{qk}$ be the weight update from Proposition 2.1. Let $t^*$ be a given token in $U$. When using $t^*$ as context, the $k$-th column of $\Delta\mathbf{W}_{qk}$ is given by*

$$\Delta\mathbf{w}_{\cdot,k}\Big|_{t_j=t^*} = [\mathbf{x}_{t^*}]_k \left( \sum_{i\in\mathcal{P}_{t^*}} \beta_{i*}\mathbf{x}_i \right), \tag{13}$$

*while the $m$-th row of $\Delta\mathbf{W}_{qk}$ is given by*

$$\Delta\mathbf{w}_{m,\cdot}\Big|_{t_j=t^*} = \left( \sum_{i\in\mathcal{P}_{t^*}} \beta_{i*}[\mathbf{x}_i]_m \right) \mathbf{x}_{t^*}. \tag{14}$$

*When predicting $t^*$, the $k$-th column of $\Delta\mathbf{W}_{qk}$ is given by*

$$\Delta\mathbf{w}_{k,\cdot}\Big|_{t_i=t^*} = \left( \sum_{j\in\mathcal{C}_{t^*}} \beta_{*j}[\mathbf{x}_j]_k \right) \mathbf{x}_{t^*}, \tag{15}$$

*while the $m$-th row of $\Delta\mathbf{W}_{qk}$ is given by*

$$\Delta\mathbf{w}_{m,\cdot}\Big|_{t_i=t^*} = [\mathbf{x}_{t^*}]_m \left( \sum_{j\in\mathcal{C}_{t^*}} \beta_{*j}\mathbf{x}_j^\top \right). \tag{16}$$

We provide proof for this proposition with related remarks in Appendix A.5.1. Note that all mathematical results derived so far rely solely on the structure of self-attention and make no assumptions about the input data. In the next section, we build on these results to link structural patterns in $\mathbf{W}_{qk}$ with the specific form of the objective function.

## 2.4. The relation between objective functions and structures in self-attention matrices

In this final section, we show how to relate these properties to the specific objective function. Crucially, the number of times a token $t^*$ appears as context or as a prediction depends on the training objective. Autoregressive training implicitly introduces directionality by predicting each token solely on its preceding tokens. In contrast, bidirectional training uses tokens as context and as predictions symmetrically. This fundamental difference affects the weight updates of $\mathbf{W}_{qk}$, and consequently, the structures encoded in its rows and columns.

We formalize this relationship in the next two theorems, under the following assumptions: (a) tokens exhibit statistical correlations, leading to partial alignment in their embeddings; (b) the entries of $\mathbf{W}_{qk}$ are i.i.d. at initialization with finite mean and variance; (c) some tokens tend to occur earlier in the sequence and are more predictive of future content; (d) bidirectional training induces approximately symmetric error signals. In this setting, we prove how the objective function influences the internal structure encoded by self-attention.

**Theorem 2.3.** *(Autoregressive training induces directionality) Let $\mathcal{V} = \{t_0, \ldots, t_V\}$ be a vocabulary of tokens, and let $\mathcal{U} \subset \mathcal{V}^N$ denote the sample space of all sequences of length $N$. Let $U = \{t_1, \ldots, t_N\} \in \mathcal{U}$ be a random variable with $U \sim P(U)$. Define $\Pr[t_j = t^*]$ as the marginal probability that the token at position $j$ equals $t^* \in \mathcal{V}$. Let $\{\mathbf{x}_1, \ldots, \mathbf{x}_N\}$ be the token embeddings corresponding to the elements of $U$, where each embedding $\mathbf{x}_i \sim \mathcal{D}$ is drawn i.i.d. from a distribution $\mathcal{D}$ with zero mean and covariance matrix $Cov(\mathbf{x}_i) = \Sigma..$ Let $\mathbf{W}_{qk}$ be query-key matrix of a self-attention mechanism, and let $\Delta\mathbf{W}_{qk}$ denote its gradient update as defined in Proposition 2.1, computed under an autoregressive objective as in Definition A.3.*

*It follows that the contribution of the token $t^*$ after training satisfies,*

$$\frac{\mathbb{E}_{\mathcal{D}}\left[\|\Delta\mathbf{w}_{\cdot,k}\|^2\right]}{\mathbb{E}_{\mathcal{D}}\left[\|\Delta\mathbf{w}_{m,\cdot}\|^2\right]} > 1 \quad \forall k, \ \forall m \ s.t. \ \Sigma_{m,m} < \frac{\text{Tr}(\Sigma)}{d}. \tag{17}$$

*Moreover, there exists a scalar $\gamma \in \mathbb{R}_{>0}$, proportional to the product of the row and column norm variances, $\gamma \propto \text{Var}(\|\mathbf{w}_{m,\cdot}\|) \cdot \text{Var}(\|\mathbf{w}_{\cdot,k}\|)$, such that for all $w > \gamma$,*

$$\Pr\left[\|\mathbf{w}_{\cdot,k}\|^2 > w\right] > \Pr\left[\|\mathbf{w}_{m,\cdot}\|^2 > w\right], \tag{18}$$

*that is, columns of $\mathbf{W}_{qk}$ are more likely than rows to have high norm under autoregressive training, indicating a directional bias in gradient updates.*

**Theorem 2.4.** *(Bidirectional training induces symmetry). Let $U = \{t_1, \ldots, t_N\}$ be a sequence of tokens. Let $\mathbf{W}_{qk}$ be the query-key matrix of a self-attention mechanism, and*

*let $\Delta\mathbf{W}_{qk}$ denote its gradient update as defined in Proposition 2.1, computed under a bidirectional objective as in Definition A.3. It follows that,*

$$\Delta\mathbf{W}_{qk} = \sum_{i=1}^{N}\sum_{j=1}^{N}\beta_{ij}\mathbf{K}_{ij}\,, \tag{19}$$

*and that every pair $(i,j)$ with $i \neq j$ contributes to the weight update with a term*

$$\Delta\mathbf{W}_{qk}\big|_{\mathbf{t}_i \leftrightarrow \mathbf{t}_j} = \beta_{ij}\mathbf{K}_{ij} + \beta_{ji}\mathbf{K}_{ij}{}^{T}\,, \tag{20}$$

*that is approximately symmetric,*

$$\Delta\mathbf{W}_{qk}\big|_{\mathbf{t}_i \leftrightarrow \mathbf{t}_j} \approx \Delta\mathbf{W}_{qk}\big|_{\mathbf{t}_i \leftrightarrow \mathbf{t}_j}^{\top}\,. \tag{21}$$

We provide a proof of these two Theorems, and the related Propositions and Lemmas in Appendix A.5.

## 3. Symmetric and directional structures are predominant in Transformer models

In this section, we validate our theoretical findings by quantifying empirically the degree of symmetry and directionality in different families of open-source Transformer models. To do so, we define two scores for symmetry and directionality in square matrices. First, we define the symmetry score $s \in \mathbb{R}$ as follows,

**Definition 3.1.** (*Symmetry score*). Given a square matrix $\mathbf{M} \in \mathbb{M}_n$ we define the symmetry score,

$$s = 2\frac{||\mathbf{M}_s||_F^2 - ||\mathbf{M}_n||_F^2}{||\mathbf{M}||_F^2}\,, \tag{22}$$

where $||\cdot||_F$ is the Frobenious norm, and $\mathbf{M}_s$ and $\mathbf{M}_n$ are the symmetric and skew-symmetric parts of the Toeplitz decomposition of $\mathbf{M}$, respectively,

$$\mathbf{M}_s = \frac{1}{2}\big(\mathbf{M} + \mathbf{M}^{\top}\big) \quad ; \quad \mathbf{M}_n = \frac{1}{2}\big(\mathbf{M} - \mathbf{M}^{\top}\big). \tag{23}$$

Here, positive and negative symmetry scores indicate the presence of symmetric and skew-symmetric structures, respectively (see Appendix A.6). Second, we define the directionality score $d \in \mathbb{R}$ as follows,

**Definition 3.2.** (*Directionality score*). Given a square matrix $\mathbf{M} \in \mathbb{M}_n$ we define the directionality score,

$$d = \frac{\bar{r}_{\mathbf{M}} - \bar{c}_{\mathbf{M}}}{\bar{r}_{\mathbf{M}} + \bar{c}_{\mathbf{M}}}\,, \tag{24}$$

where $\bar{c}_{\mathbf{M}}$ is the sum of the norm of the columns that are higher than a given threshold, as follows,

$$\bar{c}_{\mathbf{M}} = \sum_{k\in\overline{\mathcal{C}}}||\mathbf{m}_{\cdot,k}||_2 \ \text{ with } \ \overline{\mathcal{C}} = \{k \mid ||\mathbf{m}_{\cdot,k}||_2 > \mu_c + \gamma\sigma_c\}, \tag{25}$$

where $||\cdot||_2$ is the L2 norm, $||\mathbf{m}_{\cdot,k}||_2$ is the norm of the $k$-th column, $\mu_c = \mathbb{E}[||\mathbf{m}_{\cdot,k}||_2]$ and $\sigma_c = \sqrt{\mathrm{Var}||\mathbf{m}_{\cdot,k}||_2}$ are the mean and standard deviation of the column norms, $\gamma$ is a scaling factor, and similarly $\bar{r}_{\mathbf{M}}$ is the sum of the norm of the rows that are higher than a given threshold,

$$\bar{r}_{\mathbf{M}} = \sum_{k\in\overline{\mathcal{R}}}||\mathbf{m}_{k,\cdot}||_2 \ \text{ with } \ \overline{\mathcal{R}} = \{k \mid ||\mathbf{m}_{k,\cdot}||_2 > \mu_r + \gamma\sigma_r\}, \tag{26}$$

with $||\mathbf{m}_{k,\cdot}||_2$ as the norm of the $k$-th row, $\mu_r = \mathbb{E}[||\mathbf{m}_{k,\cdot}||_2]$ and $\sigma_r = \sqrt{\mathrm{Var}||\mathbf{m}_{k,\cdot}||_2}$ as the mean and standard deviation of the row norms.

Here, positive and negative directionality scores indicate the dominance of high norm rows or columns, respectively (see Appendix A.7). Finally, we compute the matrix $\mathbf{W}_{qk}$ for every layer, calculate the median symmetry and directionality score across layers, and analyze the differences between encoder- and decoder-only variants.

We find that encoder-only models remarkably show a higher degree of symmetry than decoder-only (Figure 2a). This difference is consistent across multiple families of models and input modalities, such as BERT (Devlin et al., 2019), GPT (Radford et al., 2018; 2019), LLAMA3 (Touvron et al., 2023), Phi (Hughes, 2023; Abdin et al., 2024), MISTRAL (Jiang et al., 2023), ModernBERT (Warner et al., 2024), and many others (see Figure S1 for vision and audio models). Strikingly, we observe that decoder-only models have higher degrees of directionality than encoder-only models (Figure 2b). Again, this difference is consistent across all the models and input modalities we consider. We show in Figure S2 that a similar pattern is observed when including full encoder-decoder Transformers (e.g. the language T5 models (Xue et al., 2021)), despite these models having an overall lower degree of directionality.

## 4. Experiments

In this final section, we test if using structural priors based on our previous results can improve the pretraining of Transformer models. To do so, we train Transformer models from scratch and perform a series of experiments to analyze how symmetric and directional structures develop during training across layers.

### 4.1. Evolution of symmetric and directional structures during learning

To test the applicability of our result, we first train 12-layer transformer models in both encoder and decoder modes and quantify the median symmetry and directionality scores across epochs. At initialization, the symmetry and directionality score of the matrix $\mathbf{W}_{qk}$ at any layer is zero (see Definition 3.1 and Definition 3.2 and related Appendix A.6

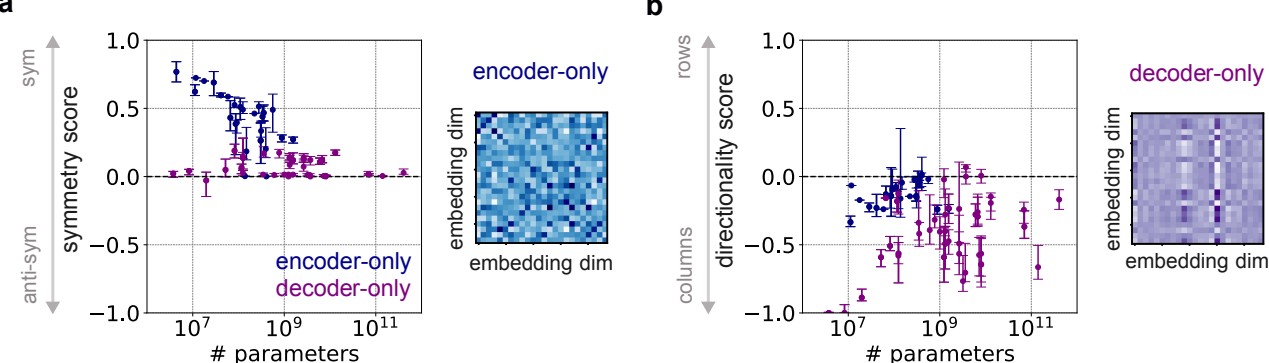

*Figure 2.* **a**) Left) Median symmetry score of the matrix $\mathbf{W}_{qk}$ as a function of the total number of parameters. Each dot corresponds to the median and the interquartile range across layers of a given pre-trained model (see Tables in Appendix 5). Right) Example of structures in the $\mathbf{W}_{qk}$ matrix of an encoder-only model (BERT Tiny, layer 1 (Turc et al., 2019)) **b**) Left) Same as in **a** for the median directionality score of the matrix $\mathbf{W}_{qk}$. Right) Example of structures in the $\mathbf{W}_{qk}$ matrix of a decoder-only model (TinyStories GPT, layer 1 (Eldan & Li, 2023))

and A.7). The incremental update of $\mathbf{W}_{qk}$ we described in the previous sections predicts that decoder-only models develop high-norm columns incrementally during training (see Theorem 2.3). Likewise, as symmetric weight updates are added to $\mathbf{W}_{qk}$ in encoder-only models, Theorem 2.4 predicts that symmetric structures emerge incrementally during training.

Consistent with our results on pre-trained models, encoder-only models show a higher degree of symmetry than decoder-only models (Figure 3a). In contrast, decoder-only models have a higher directionality score (Figure 3b). We observe this difference on all datasets we tested (Jigsaw (cjadams et al., 2017), Wikipedia (Foundation, 2022), Red Pajama (Computer, 2023), see Figure S3). Furthermore, late layers of encoder-only models are more symmetric and converge faster than early layers when training bidirectionally. At the same time, decoder-only models learn almost non-symmetric matrices with strong skew-symmetric matrices in the middle layers (Figure 3c). When training unidirectionally, both encoder and decoder models show a higher degree of directionality for late layers, which is remarkably stronger for decoder-only models (Figure 3d). We observe similar differences across layers with all the datasets we tested (Figure S4), despite these models having less significant differences in directionality scores. See Appendix B for a detailed description of the experiments.

### 4.2. Enforcing symmetry at initialization improves the training of encoder-only models

The previous section showed that symmetric structures incrementally emerge during training in the $\mathbf{W}_{qk}$ matrices of encoder-only models. Here, we first provide evidence that these findings can be exploited to speed up training using

*Table 1.* The final loss at the end of training and the speed-up for the 4, 12, and 24-layer models trained on the Jigsaw dataset (cjadams et al., 2017), Wikipedia (Foundation, 2022), and Red Pajama (Computer, 2023), with and without symmetry initialization (see Appendix B.1). Speed-up (%) is calculated by subtracting the epoch at which the symmetrically initialized model reaches the non-symmetric model's final loss from the total number of epochs, and then dividing by the total number of epochs. For example, a 50% speed-up means that the model with symmetric initialization achieves the final loss of the non-symmetric model in half the number of training epochs.

| MODEL | LOSS | SPEED-UP |
|---|---|---|
| **4-LAYER MODEL** | | |
| JIGSAW | 2.782 | |
| JIGSAW (+ SYMM) | **2.758** | 26 % |
| WIKIPEDIA | 0.984 | |
| WIKIPEDIA (+ SYMM) | **0.812** | 73 % |
| RED PAJAMA | 1.106 | |
| RED PAJAMA (+ SYMM) | **0.907** | 69 % |
| **12-LAYER MODEL** | | |
| JIGSAW | 1.419 | |
| JIGSAW (+ SYMM) | 1.430 | 0 % |
| WIKIPEDIA | 0.256 | |
| WIKIPEDIA (+ SYMM) | **0.247** | 20 % |
| RED PAJAMA | 0.297 | |
| RED PAJAMA (+ SYMM) | **0.274** | 35 % |
| **24-LAYER MODEL** | | |
| JIGSAW | 0.786 | |
| JIGSAW (+ SYMM) | **0.739** | 14% |
| WIKIPEDIA | 0.192 | |
| WIKIPEDIA (+ SYMM) | **0.166** | 44% |
| RED PAJAMA | 0.209 | |
| RED PAJAMA (+ SYMM) | **0.189** | 34% |

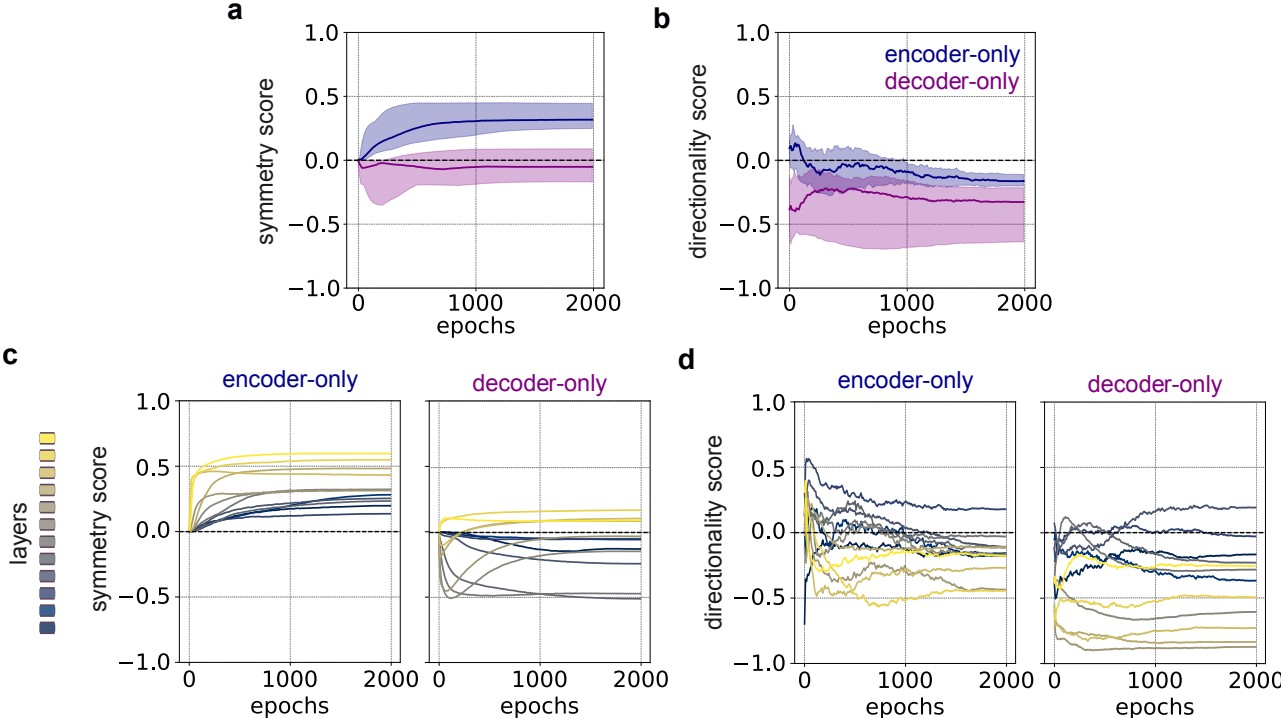

*Figure 3.* **a**) Evolution of symmetry score during training. We train a Bert-base-uncased model with 12 layers on the Wikipedia dataset (Foundation, 2022) in encoder-only (blue) and decoder-only (purple) mode, see legend. Shown are the median and the interquartile range. Shown are the median and the interquartile range. **b**) Same as in panel **a** for the median directionality score. **c**) Evolution of the median symmetry score across layers of the encoder-only (left) and decoder-only (right) models. Each layer is color-coded as shown on the legend. **d**) Same as panel **c** for the median directionality score.

symmetry as an inductive bias. Specifically, we explore how symmetric initialization influences the training dynamics of the model and whether it enhances learning efficiency and overall performance.

We train 4-layer, 12-layer, and 24-layer encoder-only models, comparing two initialization strategies: initialize the self-attention matrices independently versus initializing the $\mathbf{W}_q$ and $\mathbf{W}_k$ in each self-attention layer to ensure that $\mathbf{W}_{qk}$ is symmetric (see Appendix B). We report the results of our experiments in Table 1. We observe that enforcing symmetry at initialization leads to lower loss values at the end of training for most of the models. Importantly, symmetric initialization significantly accelerates convergence, reaching the final loss value faster than those with random initialization (up to 75% faster for 4-layer models, 35% faster for 12-layer models, and 44% faster for 24-layer models; see also Figure S5a). Moreover, we observe that self-attention matrices initialized symmetrically lose symmetry during training but converge to higher symmetry levels than random initialization (Figure S5b) This symmetric initialization decreases the gap in symmetric scores between layers compared to random initialization (Figure S5c). When we tested symmetric initialization on vision Transformers, the

improvements were not significant compared to the gains observed in language models (see Appendix C for a detailed analysis). These results highlight that embedding symmetry as an inductive bias across all Transformer layers can enhance training efficiency and model performance.

## 5. Related work

**Mechanistic Interpretability (MI).** In contrast to interpretability approaches that focus on explaining specific data instances by analyzing features (Wu et al., 2020; Lundstrom et al., 2022), attention scores (Hoover et al., 2020; Barkan et al., 2021; Yeh et al., 2023), or output variations (Jin et al., 2020; Wang et al., 2022a), mechanistic interpretability (MI) seeks to provide a more general understanding of Transformer models. MI is based on the study of "circuits," analyzing the between activations across different components of a Transformer (Olah et al., 2020). Following the categorization by (Rai et al., 2024), MI techniques include: (i) The *logit lens* (nostalgebraist, 2020; Geva et al., 2021) projects layer activations or weights into the vocabulary space $\mathcal{V}$, allowing the derivation of logits and revealing the influence of individual components on the prediction. This

technique can also be applied to query-key matrices $\mathbf{W}_{qk}$ to study how attention heads transform source into target tokens (Dar et al., 2023); (ii) *Probing techniques* allow to identify correlations between layer activations and features by training a linear classifier or shallow neural network (the probe) to predict the presence of a feature in layer activations (Dalvi et al., 2019; Gurnee et al., 2023); (iii) *Sparse autoencoders* map activations into a higher-dimensional yet sparse representation, facilitating the identification of independent (monosemantic) features (Huben et al., 2024; Bricken et al., 2023); (iv) *Visualization techniques* facilitate the analysis of attention scores (Olsson et al., 2022; Lieberum et al., 2023) and neuronal activity patterns (Elhage et al., 2022) by rendering them in a graphical format. However, their utility often depends on human comprehension of the resulting visualizations; (v) *Ablation studies* assess the importance of model components by systematically removing or modifying them and observing the resulting behavioral changes (Olsson et al., 2022; Wang et al., 2022b); (vi) *Causal mediation analysis (CMA)* analyzes the importance of components (Vig et al., 2020; Meng et al., 2022) or connections (Wang et al., 2022b; Goldowsky-Dill et al., 2023) between them by introducing perturbations (e.g., noise) and selectively patching them to measure the recovery of lost capabilities. Our work adds a new perspective on MI by providing a scalable and generalizable approach to the mechanistic understanding of self-attention. In contrast to existing work, it is not limited to analyzing fully trained models but investigates the influence of learning and can analyze models of different sizes across all modalities.

**MI for model enhancement.** Insights from MI have been instrumental in various applications, including improving model safety (Belrose et al., 2023), updating the model's learned knowledge (Meng et al., 2022), and guiding the generation process of generative models (Geva et al., 2022). One of the most exciting applications is leveraging MI techniques to improve model performance. For instance, Skean et al. (2025) conduct information-theoretic and geometric analyses to show that intermediate layers exhibit a favorable trade-off between compression and signal preservation. Leveraging this insight, they demonstrate that features extracted from these layers consistently outperform final-layer representations across a broad range of downstream tasks. Similarly, Trockman & Kolter (2023) observe that query-key matrices ($\mathbf{W}_{qk}$) frequently exhibit a pronounced negative diagonal, prompting them to initialize it with approximately the identity matrix, leading to enhanced accuracy in image classification tasks. Similar to (Trockman & Kolter, 2023), we demonstrate that findings about the structure of $\mathbf{W}_{qk}$ can inform initialization strategies that improve Transformer performance. However, since the identity matrix is one instance of a symmetric matrix, we consider the work by (Trockman & Kolter, 2023) as a special instance of our broader approach, confirming our findings in the image domain.

# 6. Discussion

In this work, we demonstrate how bidirectional and autoregressive objective functions influence the structure of the query-key matrix $\mathbf{W}_{qk}$ in self-attention, enhancing our understanding of Transformer models. Our mathematical framework shows that bidirectional training induces symmetric structures in $\mathbf{W}_{qk}$, whereas autoregressive training results in matrices characterized by directionality and column dominance. To empirically validate our analysis, we develop and apply symmetry and directionality scores to various Transformer encoder and decoder models across multiple modalities, including text, audio, and images. Our results reveal that bidirectionally trained encoder models exhibit high symmetry, while autoregressively trained decoder models demonstrate strong directionality, thereby supporting the predictions of our mathematical framework. This suggests that self-attention inherently reflects these structural properties, contributing to the mechanistic interpretability of Transformer models. Finally, we leverage our findings to improve convergence speed during bidirectional training by initializing $\mathbf{W}_q$ and $\mathbf{W}_k$ matrices such that $\mathbf{W}_{qk}$ is symmetric. While our findings mark an initial step toward leveraging symmetry for more efficient Transformer training, further research is required to assess the scalability of symmetric initialization in large-scale models and across diverse domains. Furthermore, it is important to explore strategies for leveraging the directionality structures of decoder-only models. For instance, incorporating structural constraints into the objective function or weight regularization could enhance training efficiency and stability in autoregressive settings.

Intriguingly, we found that there are similarities between the structure of the attention weights that we find and existing connectivity patterns in cortex, as areas dedicated and image comprehension show symmetric connectivity patterns (Song et al., 2005; Wang et al., 2006), while areas dedicated to language are highly directional (Peng et al., 2024). Further studies could use those patterns to connect self-attention mechanisms and models of information processing in the brain (Gershman et al., 2025).

By bridging theoretical insights with practical improvements, our work not only advances the interpretability of self-attention but also provides a foundation for optimizing Transformer architectures. Ultimately, these findings contribute to a deeper understanding of the mechanisms governing self-attention, paving the way for more reliable and efficient Transformer-based models.

## Authorship contributions

Conceptualization: MS, TS, BG. Mathematical analysis: MS, PVA. Analysis of pre-trained models: MS, PS. Experiments: PS. Writing: MS, PS, BG. Supervision: TS, BG.

## Code availability

The code to reproduce our results and all the Figures in the main text and Supplementary Materials is freely available at github.com/matteosaponati/attention-geometry.

## Declaration of Interests

The authors declare no competing interests.

## Acknowledgements

MS was supported by an ETH Postdoctoral fellowship (reference nr. 23-2 FEL-042). PVA was supported by a UZH Postdoctoral fellowship (reference nr. K-76207-01-01). TS and PS were supported by DIZH Fellowship "Stability of self-organizing net fragments as inductive bias for next-generation deep learning" from the ZHAW. This work was supported as part of the Swiss AI Initiative by a grant from the Swiss National Supercomputing Centre (CSCS) under project ID a06 on Alps. We thank Yassine Taoudi Benchekroun, the Grewe lab, Alexander Nedergaard, Thomas Hofmann, and the Data Analytics lab for their helpful comments and insightful discussions.

## Impact Statement

This paper presents work whose goal is to advance the field of Machine Learning. There are many potential societal consequences of our work, none which we feel must be specifically highlighted here.

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

# A. Mathematical proofs

## A.1. Preliminaries

Following the notation in (Vaswani et al., 2017; Radford et al., 2018), we define a Transformer architecture as,

**Definition A.1.** (***Transformer architecture***). Let $\mathbf{U} \in \mathbb{R}^{N,V}$ be a matrix representing the sequence of $N$ one-hot encoded tokens of dimension $V$. A Transformer architecture consists of $L$ stacked attention blocks, each one composed of a self-attention layer and a feedforward layer, as follows,

$$\begin{cases} \mathbf{X}_0(\mathbf{U}) = \mathbf{U}\mathbf{W_e} + \mathbf{W}_p \\ \begin{cases} \hat{\mathbf{X}}_l = \mathbf{X}_{l-1} + a_l(\mathbf{X}_{l-1}; \mathbf{W}_q^l, \mathbf{W}_k^l, \mathbf{W}_v^l) \\ \mathbf{X}_l = \hat{\mathbf{X}}_l + m_l(\hat{\mathbf{X}}_l; \mathbf{W}_1^l, \mathbf{W}_2^l) \end{cases} \forall l \in [1, L] \\ \sigma(\mathbf{Z}) = \sigma(\mathbf{X}_L \mathbf{W_u}), \end{cases} \tag{S1}$$

where $\mathbf{W_e} \in \mathbb{R}^{V,d}$ represents the linear transformation from the vocabulary space to the embedding space of dimension $d$, $\mathbf{W_p} \in \mathbb{R}^{V,d}$ represents the positional encoding, $\mathbf{X}_0 \in \mathbb{R}^{N,d}$ is the initial embedding of the sequence, $a_l(\cdot)$ is a self-attention function given by

$$a(\mathbf{X}_{l-1}) = \mathbf{A}^l(\mathbf{X}_{l-1})\mathbf{V}^l(\mathbf{X}_{l-1}) \tag{S2}$$

where the matrix of attention scores $\mathbf{A}^l(\mathbf{X}_{l-1})$ is given by

$$\begin{cases} \mathbf{Q}^l(\mathbf{X}_{l-1}) = \mathbf{X}_{l-1}\mathbf{W}_q^l \\ \mathbf{K}^l(\mathbf{X}_{l-1}) = \mathbf{X}_{l-1}\mathbf{W}_k^l \\ \mathbf{A}^l(\mathbf{X}_{l-1}) = \sigma\left(\frac{1}{\sqrt{d}}\mathbf{Q}^l\mathbf{K}^{l^T}\right), \end{cases} \tag{S3}$$

where $1/\sqrt{d}$ is a constant normalization factor, and $\mathbf{W}_q^l \in \mathbb{R}^{d,d}$ and $\mathbf{W}_k^l \in \mathbb{R}^{d,d}$ represent linear transformations within the embedding space, $\mathbf{V}^l(\mathbf{X}_{l-1}) = \mathbf{X}_{l-1}\mathbf{W}_v^l$ represents a linear transformation within the embedding space, $m_l(\cdot)$ is a position-wise feedforward layer with hidden dimension $d_f$ and learnable matrices $\mathbf{W}_1^l \in \mathbf{R}^{d,d_f}$ and $\mathbf{W}_2^l \in \mathbf{R}^{d_f,d}$, $\mathbf{W_u} \in \mathbb{R}^{d,V}$ represents the linear transformation from the embedding space back to the vocabulary space, $\sigma(\cdot)$ is the row-wise softmax function, and $\sigma(\mathbf{Z}) \in \mathbb{R}^{N,V}$ is the estimated probability distribution over the vocabulary. We omit layer normalization and biases for simplicity (see also (Elhage et al., 2021)).

Furthermore, we use the following definition of a bilinear form,

**Definition A.2.** (***Bilinear form***). A bilinear form on a vector space $V$ over a field $F$ is a map $M : V \times V \to F$ that is linear in each argument separately, that is,

$$\begin{aligned} M(a\mathbf{x} + b\mathbf{y}, \mathbf{z}) &= aM(\mathbf{x}, \mathbf{z}) + bM(\mathbf{y}, \mathbf{z}) \\ M(\mathbf{x}, a\mathbf{y} + b\mathbf{z}) &= aM(\mathbf{x}, \mathbf{y}) + bM(\mathbf{x}, \mathbf{z}), \end{aligned} \tag{S4}$$

for all $\mathbf{x}, \mathbf{y}, \mathbf{z} \in V$ and $a, b \in F$. Let $\{\mathbf{e}_1, \ldots, \mathbf{e}_d\}$ be a basis for the vector space $V$. The matrix $\mathbf{M}$ such that $[\mathbf{M}]_{ij} = M(\mathbf{e}_i, \mathbf{e}_j)$ is the matrix of the bilinear form on this basis, and it follows

$$M(\mathbf{x}, \mathbf{y}) = \mathbf{x}^\top \mathbf{M}\mathbf{y} \tag{S5}$$

Finally, we provide the following definition of autoregressive and bidirectional training objectives,

**Definition A.3.** (***Autoregressive and bidirectional objectives***). Let $U = \{t_1, \ldots, t_N\}$ a sequence of tokens. The joint probability of $U$ is factorized autoregressively as follows,

$$\Pr[U] = \Pr[t_1, \ldots, t_N] = \Pi_{i=1}^N \Pr[t_i | t_1, \ldots, t_{i-1}]. \tag{S6}$$

During autoregressive training, a model with a set of parameters $\mathcal{W}$ is optimized to minimize the following negative log-likelihood

$$\mathcal{L}(U; \mathcal{W}) = -\sum_{i=1}^N \log p_{\mathcal{W}}(t_i | \{t_j\} : j < i), \tag{S7}$$

where the conditional probabilities are modeled with learnable parameters $\mathcal{W}$. During bidirectional training, a model with a set of parameters $\mathcal{W}$ is optimized to minimize the following negative log-likelihood

$$\mathcal{L}(U; \mathcal{W}) = -\sum_{i=1}^{N} \log p_{\mathcal{W}}(t_i \,|\, \{t_j\} : j \neq i). \tag{S8}$$

In practice, only a subset $M \in [1, \ldots, N]$ of the tokens are predicted as in Masked Language Modelling (MLM) (see Devlin et al., 2019; Warner et al., 2024), leading to the following negative log-likelihood

$$\mathcal{L}(U; \mathcal{W}) = -\sum_{i \in M} \log p_{\mathcal{W}}(t_i \,|\, \{t_j\} : j \notin M). \tag{S9}$$

### A.2. Related remarks to Section 2.1

*Remark* A.4. Each element of the sum in Equation (4) implicitly defines an operator in the subspace spanned by $\mathbf{X} = \{\mathbf{x}_0^\top, \mathbf{x}_1^\top, \ldots, \mathbf{x}_N^\top\}$ given the transformed embedding space defined by $\mathbf{W}_{qk}$,

$$\sum_j \alpha_{ij}(\mathbf{W}_{qk}) \, \mathbf{x}_j = \sum_j \mathbf{x}_i^\top \mathbf{W}_{qk} \mathbf{x}_j \, \mathbf{x}_j = \sum_j P_{\mathbf{W}_{qk}}(\mathbf{x}_i, \mathbf{x}_j), \tag{S10}$$

where $P_{\mathbf{W}_{qk}}(\cdot, \mathbf{x}_j)$ are operators over the subset $\mathbf{X}$. The set $\mathbf{X}$ is in general linearly dependent, and the number of tokens $N$ in the sequence differs from the embedding space dimension $d$. Furthermore, $\mathbf{W}_{qk}$ represents a general bilinear form (see Definition A.2), which may not satisfy all the defining axioms of a formal inner product — namely, linearity, conjugate symmetry, and positive definiteness. Finally, the operators $P_{\mathbf{W}_{qk}}(\cdot \mathbf{x}_j)$ are not formal projection operators since they are not nilpotent ($P_{\mathbf{W}_{qk}}(\cdot \mathbf{x}_j) \circ P_{\mathbf{W}_{qk}}(\cdot \mathbf{x}_j) \neq P_{\mathbf{W}_{qk}}(\cdot \mathbf{x}_j)$). Nonetheless, the bilinear map $\mathbf{W}_{qk} : \mathbb{R}^d \times \mathbb{R}^d \to \mathbb{R}$ still associates any pair of vectors with a scalar value quantifying their alignment as determined by the geometric relations encoded in $\mathbf{W}_{qk}$. Therefore, self-attention computes a generalized decomposition of $\mathbf{x}_i$ on $\mathrm{Conv}(\mathbf{X})$ in the transformed embedding space defined by $\mathbf{W}_{qk}$. A convex combination ensures that the resulting vector remains within the region enclosed by the basis vectors $\mathbf{X}$.

*Remark* A.5. Following Definition A.1, multi-head attention consists of parallelizing the self-attention operation across $H$ different heads with an embedding space $d_h < d$,

$$\hat{\mathbf{X}}(\mathbf{X}) = \mathbf{X} + \mathrm{concat}(\mathbf{A}_1 \mathbf{V}_1, \mathbf{A}_2 \mathbf{V}_2, \ldots, \mathbf{A}_h \mathbf{V}_h) \mathbf{W}_o \tag{S11}$$

where $\mathbf{A}_h = \sigma(d^{-1/2} \, \mathbf{X} \, \mathbf{W}_{q,h} \, \mathbf{W}_{k,h}^\top \, \mathbf{X}^T)$ is the self-attention of the $h$-th head, $\mathbf{W}_{q,h} \in \mathbb{R}^{d,d_h}$, $\mathbf{W}_{k,h} \in \mathbb{R}^{d,d_h}$ and $\mathbf{W}_{v,h} \in \mathbb{R}^{d,d_h}$ are the query, key, and value matrices of the $h$-th attention head, respectively, and $\mathbf{W}_o \in \mathbb{R}^{d,d}$ is a linear transformation (Vaswani et al., 2017). Operationally, the self-attention computation is performed in parallel by factorizing the $\mathbf{W}_q$ and $\mathbf{W}_k$ matrices into $H$ rectangular blocks, as follows,

$$\begin{aligned}
\mathbf{W}_q &= \big[\mathbf{W}_{q,1} \big| \mathbf{W}_{q,2} \big| \ldots \big| \mathbf{W}_{q,H}\big] \\
\mathbf{W}_k &= \big[\mathbf{W}_{k,1} \big| \mathbf{W}_{k,2} \big| \ldots \big| \mathbf{W}_{k,H}\big],
\end{aligned} \tag{S12}$$

and performing the matrix multiplication $\mathbf{W}_{q,h} \mathbf{W}_{k,h}^\top$ per every $h$-th head independently in one step. It follows that the full $\mathbf{W}_{qk}$ matrix is given by the sum of the bilinear forms $\mathbf{W}_{qk,h}$ of every head, as follows,

$$\mathbf{W}_{qk} = \mathbf{W}_q \mathbf{W}_k^\top = \sum_h \mathbf{W}_{q,h} \mathbf{W}_{k,h}^\top = \sum_h \mathbf{W}_{qk,h} \tag{S13}$$

where each $\mathbf{W}_{qk,h} \in \mathbb{R}^{d,d}$ is a square matrix with $\mathrm{rank}(\mathbf{W}_{qk,h}) \leq d_h$. Therefore, each head perform independent projections onto $\mathrm{Conv}(\mathbf{X})$ that are then summed together, as follows,

$$\sum_j \hat{\alpha}_{ij} \, \mathbf{x}_j = \sum_j \mathbf{x}_i^\top \mathbf{W}_{qk} \mathbf{x}_j \, \mathbf{x}_j = \sum_j \mathbf{x}_i^\top \Big(\sum_h \mathbf{W}_{qk,h}\Big) \mathbf{x}_j \, \mathbf{x}_j = \sum_j \sum_h \langle \mathbf{x}_i, \mathbf{W}_{qk,h} \mathbf{x}_j \rangle \, \mathbf{x}_j, \tag{S14}$$

thus performing the same operations as in Equation (4).

## A.3. Related remarks to Section 2.2

Let $\mathcal{V} = \{t_1, \dots, t_V\}$ be a vocabulary of tokens, and let $D$ be a dataset of sequences $U_u \in \mathcal{V}^N$ of length $N$, where $D = \{U_u = (x_1^u, \dots, x_N^u)\}_{u=1}^D \subset \mathcal{V}^N$. For a single sequence $U_u$ with joint distribution $p(U_u)$, the chain rule implies that for any permutation $\sigma$ of $\{1, \dots, N\}$, the joint probability can be written as,

$$p(U_u) = \prod_{t=1}^N p\big(x_{\sigma(t)}^u \mid x_{\sigma(1)}^u, \dots, x_{\sigma(t-1)}^u\big). \tag{S15}$$

Given a set of conditional parents $\mathcal{D}_i \subset \{1, \dots, N\} \setminus \{i\}$, define the **local negative log-likelihood** as:

$$\mathcal{L}(U; \mathcal{W}) = -\sum_{i=1}^N \log p_{\mathcal{W}}(x_i \mid x_{\mathcal{D}_i}), \tag{S16}$$

where $\mathcal{W}$ denotes the model parameters. Let $G$ be a directed graph where each edge is $\mathcal{D}_i \to i$. We distinguish two cases:

1. If $G$ is a directed acyclic graph (DAG), then a topological ordering $\tau$ exists such that $\mathcal{D}_{\tau(i)} \subseteq \{\tau(1), \dots, \tau(i-1)\}$. In this case, the joint distribution $p_{\mathcal{W}}(x) = \prod_{i=1}^N p_{\mathcal{W}}(x_{\tau(i)} \mid x_{\mathcal{D}_{\tau(i)}})$ is properly normalized and the local negative log-likelihood in (S16) is equal to the true negative log-likelihood. For example, during autoregressive training, each token is conditioned only on its predecessors,

$$\mathcal{L}(U; \mathcal{W}) = -\sum_{i=1}^N \log p_{\mathcal{W}}(x_i \mid x_1, \dots, x_{i-1}) = -\sum_{i=1}^N \log p_{\mathcal{W}}(x_i \mid x_{<i}). \tag{S17}$$

2. If $G$ contains directed cycles, the product $\prod_{i=1}^N p_{\mathcal{W}}(x_i \mid x_{\mathcal{D}_i})$ is not guaranteed to be normalized, and thus does not define a proper joint likelihood. In this case, one can instead minimize the following pseudo log-likelihood,

$$\hat{\mathcal{L}}(U; \mathcal{W}) = -\sum_{i=1}^N \log p_{\mathcal{W}}(x_i \mid x_1, \dots, x_{i-1}, x_{i+1}, \dots, x_N) = -\sum_{i=1}^N \log p_{\mathcal{W}}(x_i \mid x_{-i}), \tag{S18}$$

where $x_{-i}$ denotes all tokens in the sequence except $x_i$. Following Besag's theorem (Besag, 1975), the maximum pseudo-likelihood estimator is consistent under standard regularity conditions, and (S18) remains compatible with stochastic-gradient optimization even though it is not the true negative log-likelihood (Yang et al., 2019).

## A.4. Proof of Proposition 2.1 and related remarks

*Proof.* Let $U = \{t_1, \dots, t_N\}$ be a sequence of $N$ tokens. Let $\mathcal{L}(U; \mathcal{W})$ be the negative log-likelihood of each token $t_i$, expressed as

$$\mathcal{L}(U; \mathcal{W}) = \sum_{i=1}^N \mathcal{L}(t_i) = -\sum_{i=1}^N \log p_{\mathcal{W}}(t_i \mid t_{\mathcal{D}_i}), \tag{S19}$$

where $\mathcal{D}_i \subset [0, 1, \dots, N]$ is the conditioning set, that is, the set of indices defining the set of tokens $\{t_j : j \in \mathcal{D}_i\}$ of the conditional probability distribution. Let $\mathbf{U} = [\mathbf{t}_0, \mathbf{t}_1, \dots, \mathbf{t}_N]$ be the sequence of $N$ one-hot encoded tokens $\mathbf{t}_i \in \mathbb{R}^V$ associated with $U$, where $V$ is the dimension of the vocabulary. Let $\mathcal{L}(\mathbf{t}_i)$ be the cross-entropy of the one-hot encoded token $\mathbf{t}_i$ and the estimated probability distribution $\sigma(\mathbf{z}_i) \in \mathbb{R}^V$, as follows,

$$\mathcal{L}(\mathbf{U}) = \sum_{i=1}^N \mathcal{L}(\mathbf{t}_i) = \sum_{i=1}^N \mathbf{t}_i \log(\sigma(\mathbf{z}_i)), \tag{S20}$$

where we let $\mathbf{z}_i$ be the prediction of the $i$-th token $\mathbf{t}_i$ from the representations in the last layer of a Transformer model, following Definition A.1,

$$\begin{cases} \mathbf{x}_i^{0\top} = \mathbf{t_i}^\top \mathbf{W_e} + \mathbf{W}_p \\ \mathbf{x}_i^{l\top} = \mathcal{F}_l(\mathbf{x}_i^{l-1}) \quad \forall l \in [1, L] \\ \sigma(\mathbf{z}_i) = \sigma\big(\mathbf{x}_i^{L\top} \mathbf{W_u}\big), \end{cases} \tag{S21}$$

where $\mathbf{x}_i^{l\top} = \mathcal{F}_l(\mathbf{x}_i^{l-1})$ is a short notation for the self-attention and multi-layered perception transformation of the $l$-th layer,

$$\mathcal{F}_l(\mathbf{x}_i^{l-1}) = \begin{cases} \hat{\mathbf{x}}_i^{l\top} = \hat{\mathbf{x}}_i^{l-1\top} + a_l(\mathbf{x}_i^{l-1}) \\ \mathbf{x}_i^{l\top} = \hat{\mathbf{x}}_i^{l\top} + m_l(\hat{\mathbf{x}}_i^l) \end{cases} , \tag{S22}$$

where the self-attention function is given by

$$a_l(\mathbf{x}_i^{l-1}) = \sum_{j \in \mathcal{D}_i} \alpha_{ij}^l(w^l) \mathbf{x}_j^{l-1\top} \mathbf{W}_v^l . \tag{S23}$$

Let attention coefficients $\alpha_{ij}^l \equiv \alpha_{ij}^l(w^l)$ of the $l$-th layer be parameterized with a general parameter $w^l$. Let the gradient of $\mathcal{L}(\mathbf{t}_i)$ w.r.t. $w_l$ (the parameterization of the attention scores) be factorized as follows,

$$\nabla_{w^l} \mathcal{L}(\mathbf{t}_i) = \frac{\partial \mathcal{L}(\mathbf{t}_i)}{\partial \alpha_{ij}^l} \frac{\partial \alpha_{ij}^l}{\partial w^l} . \tag{S24}$$

It follows that,

$$\begin{aligned} \frac{\partial \mathcal{L}(\mathbf{t}_i)}{\partial \alpha_{ij}^l} &= \frac{\partial \mathcal{L}(\mathbf{t}_i)}{\partial \mathbf{z}_i} \frac{\partial \mathbf{z}_i}{\partial \mathbf{x}_i^L} \frac{\partial \mathbf{x}_i^L}{\partial \hat{\mathbf{x}}_i^l} \frac{\partial \hat{\mathbf{x}}_i^l}{\partial \alpha_{ij}^l} \\ &= (\mathbf{t}_i - \sigma(\mathbf{z}_i))^\top \mathbf{W}_u^\top \frac{\partial \mathbf{x}_i^L}{\partial \hat{\mathbf{x}}_i^l} \mathbf{W}_v^{l\top} \sum_{j \in \mathcal{D}_i} \mathbf{x}_j^{l-1} , \end{aligned} \tag{S25}$$

where the term $\partial \mathbf{x}_i^L / \partial \hat{\mathbf{x}}_i^l$ includes the set of partial derivatives that define the gradient of the representation $\mathbf{x}_i^L$ at the last layer w.r.t. the self-attention representation $\hat{\mathbf{x}}_i^l$ at the $l$-layer, as follows,

$$\frac{\partial \mathbf{x}_i^L}{\partial \hat{\mathbf{x}}_i^l} = \left( 1 + \sum_{m=l}^{L-1} \mathcal{F}_l'(\mathbf{x}_i^m) \right) \left( 1 + m_l'(\hat{\mathbf{x}}_i^l) \right) , \tag{S26}$$

where

$$\mathcal{F}_l'(\mathbf{x}_i^m) = \frac{\partial}{\partial \mathbf{x}_i^l} \mathcal{F}_l(\mathbf{x}_i^m) \quad ; \quad m_l'(\hat{\mathbf{x}}_i^l) = \frac{\partial}{\partial \hat{\mathbf{x}}_i^l} m_l(\hat{\mathbf{x}}_i^l) . \tag{S27}$$

Let $\delta_i^l$ be the error at the last layer propagated to the self-attention function at the $l$-th layer, as follows,

$$\delta_i^{l\top} = (\mathbf{t}_i - \sigma(\mathbf{z}_i))^\top \mathbf{W}_u^\top \frac{\partial \mathbf{x}_i^L}{\partial \hat{\mathbf{x}}_i^l} \mathbf{W}_v^{l\top} , \tag{S28}$$

thus obtaining the following equation for the gradient,

$$\nabla_{w^l} \mathcal{L}(\mathbf{t}_i) = \delta_i^{l\top} \sum_{j \in \mathcal{D}_i} \mathbf{x}_j^{l-1} \frac{\partial \alpha_{ij}^l}{\partial w^l} . \tag{S29}$$

Let the attention scores be computed without the row-wise softmax operation and explicitly with the bilinear form $\mathbf{W}_{qk}$, such that the following expression gives the score between the $i$-th and $j$th token,

$$\alpha_{ij}^l \equiv \alpha_{ij}^l(\mathbf{W}_{qk}^l) = \mathbf{x}_i^{l-1\top} \mathbf{W}_{qk}^l \mathbf{x}_j^{l-1} , \tag{S30}$$

from which we obtain

$$\frac{\partial \alpha_{ij}^l}{\partial \mathbf{W}_{qk}^l} = \mathbf{x}_i^{l-1} \mathbf{x}_j^{l-1\top} = \mathbf{K}_{ij}^{l-1} , \tag{S31}$$

where $\mathbf{K}_{ij}^{l-1} \in \mathbb{M}_n$ is a square rank-1 matrix given by the outer product between the $i$-th and $j$-th token from the $l-1$-th layer. It follows that the total gradient of $\mathcal{L}(\mathbf{t}_i)$ is given by

$$\nabla_{\mathbf{W}_{qk}^l} \mathcal{L}(\mathbf{t}_i) = \delta_i^{l\top} \sum_{j \in \mathcal{D}_i} \mathbf{x}_j^{l-1} \mathbf{K}_{ij}^{l-1} , \tag{S32}$$

where we notice that, for every $j$, the term $\delta_i^{l\top} \mathbf{x}_j^{l-1}$ is a scalar quantity that we define as $\beta_{ij}^l$, thus obtaining,

$$\nabla_{\mathbf{W}_{qk}^l} \mathcal{L}(\mathbf{t}_i) = \sum_{j \in \mathcal{D}_i} \beta_{ij}^l \mathbf{K}_{ij}^{l-1} , \tag{S33}$$

and therefore,

$$\nabla_{\mathbf{W}_{qk}^l} \mathcal{L}(U) = \sum_i \sum_{j \in \mathcal{D}_i} \beta_{ij}^l \mathbf{K}_{ij}^{l-1} \quad \forall \mathbf{W}_{qk}^l, \, l \in [1, L] . \tag{S34}$$

We can rewrite the double summation either as,

$$\nabla_{\mathbf{W}_{qk}^l} \mathcal{L}(U) = \sum_i \sum_{j \in \mathcal{C}_i} \beta_{ij}^l \mathbf{K}_{ij}^{l-1} \quad \forall \mathbf{W}_{qk}^l, \, l \in [1, L] . \tag{S35}$$

where $\mathcal{C}_j \subset [0, 1, \ldots, N]$ is the set of indices defining the set of tokens $\{t_j\}$ used to predict $t_i$, or equivalently as,

$$\nabla_{\mathbf{W}_{qk}^l} \mathcal{L}(U) = \sum_{i \in \mathcal{P}_j} \sum_j \beta_{ij}^l \mathbf{K}_{ij}^{l-1} \quad \forall \mathbf{W}_{qk}^l, \, l \in [1, L] , \tag{S36}$$

where $\mathcal{P}_j \subset [0, 1, \ldots, N]$ is the set of indices defining the set of tokens $\{t_i\}$ that are predicted by $t_j$, thus concluding the proof. $\qquad \square$

In standard Transformer models, the bilinear form $\mathbf{W}_{qk}$ is not directly computed, and as such, it is not explicitly updated through gradient descent. Nonetheless, $\mathbf{W}_{qk}$ is implicitly updated with a combination of the weight updates of $\mathbf{W}_q$ and $\mathbf{W}_k$ having the same form as in Proposition 2.1, see the following.

*Remark* A.6. Let $\mathcal{L}(\mathbf{U})$ be the negative log-likelihood of a given sequence of one-hot encoded tokens $\mathbf{U}$. Let $\mathbf{W}_q^l$ and $\mathbf{W}_k^l$ be the query and key transformation matrices of the $l$-th layer of Transformer models, see Definition A.1. Let $\mathbf{W}_q^l$ and $\mathbf{W}_k^l$ be updated via gradient descent, that is, $\mathbf{W}_q^l \to \mathbf{W}_q^l + \eta \nabla_{\mathbf{W}_q^l} \mathcal{L}(\mathbf{U})$ and $\mathbf{W}_k^l \to \mathbf{W}_k^l + \eta \nabla_{\mathbf{W}_k^l} \mathcal{L}(\mathbf{U})$, where $\eta$ is the learning rate. It follows that the matrix $\mathbf{W}_{qk}^l = \mathbf{W}_q^l \mathbf{W}_k^{l\top}$ is implicitly updated following,

$$\begin{aligned} \mathbf{W}_{qk}^l &\to \left( \mathbf{W}_q^l + \eta \nabla_{\mathbf{W}_q^l} \mathcal{L}(U) \right) \left( \mathbf{W}_k^l + \eta \nabla_{\mathbf{W}_k^l} \mathcal{L}(U) \right)^\top \\ &= \mathbf{W}_q^l \mathbf{W}_k^{l\top} + \eta \left( \mathbf{W}_q^l \nabla_{\mathbf{W}_k^l} \mathcal{L}^\top(U) + \nabla_{\mathbf{W}_q^l} \mathcal{L}(U) \mathbf{W}_k^{l\top} \right) + o(\eta^2) \\ &= \mathbf{W}_{qk}^l + \eta \sum_i \sum_{j \in C_i} \beta_{ij}^l \left( \mathbf{W}_q^l \mathbf{W}_q^{l\top} \mathbf{K}_{ij}^{l-1} + \mathbf{K}_{ij}^{l-1} \mathbf{W}_k^l \mathbf{W}_k^{l\top} \right) + o(\eta^2) \\ &= \mathbf{W}_{qk}^l + \Delta \mathbf{W}_{qk}^l + o(\eta^2) \simeq \mathbf{W}_{qk}^l + \Delta \mathbf{W}_{qk}^l , \end{aligned} \tag{S37}$$

assuming that the learning rate $\eta$ is small. Therefore, the implicit weight update of $\mathbf{W}_{qk}^l$ following gradient descent is given by

$$\Delta \mathbf{W}_{qk}^l = \eta \sum_i \sum_{j \in C_i} \beta_{ij}^l \left[ \left( \mathbf{W}_q^l \mathbf{W}_q^{l\top} \mathbf{K}_{ij}^{l-1} \right) + \left( \mathbf{K}_{ij}^{l-1} \mathbf{W}_k^l \mathbf{W}_k^{l\top} \right) \right] \propto \sum_i \sum_{j \in C_i} \beta_{ij}^l \mathbf{K}_{ij}^{l-1} , \tag{S38}$$

where both $\mathbf{W}_q^l \mathbf{W}_q^{l\top} \mathbf{K}_{ij}^{l-1}$ and $\mathbf{K}_{ij}^{l-1} \mathbf{W}_k^l \mathbf{W}_k^{l\top}$ are rank-1 matrices,

$$\begin{aligned} \mathbf{W}_q^l \mathbf{W}_q^{l\top} \mathbf{K}_{ij}^{l-1} &= \left( \mathbf{W}_q^l \mathbf{W}_q^l \mathbf{x}_i^{l-1} \right) \mathbf{x}_j^{l-1\top} = \bar{\mathbf{x}}_i^{l-1} \mathbf{x}_j^{l-1\top} \\ \mathbf{K}_{ij}^{l-1} \mathbf{W}_k^l \mathbf{W}_k^{l\top} &= \mathbf{x}_i^{l-1} \left( \mathbf{x}_j^{l-1\top} \mathbf{W}_k^l \mathbf{W}_k^{l\top} \right) = \mathbf{x}_i^{l-1} \bar{\mathbf{x}}_j^{l-1\top} . \end{aligned} \tag{S39}$$

## A.5. Formal proofs of Theorem 2.3 and 2.4: the connection between objective functions, directionality, and symmetry

In this section, we provide formal proofs of Theorems 2.3 and 2.4, which characterize the directional and symmetric structures that emerge during autoregressive and bidirectional training, respectively. To this end, we introduce a series of intermediate Propositions and Lemmas, following these steps:

1. We begin by showing that when $t^*$ is used as a context token to predict other tokens, all predicted tokens contribute to the *column space* of $\mathbf{W}_{qk}$, while only $t^*$ contributes to the *row space*. Conversely, when $t^*$ is the token being predicted, only $t^*$ contributes to the column space, whereas all context tokens contribute to the row space (Section A.5.1).

2. Next, we show that under reasonable assumptions about the statistical distribution of token embeddings, these structural properties imply that the expected norm of column updates exceeds that of row updates when $t^*$ is used as context. In contrast, row updates dominate when $t^*$ is the predicted token (Section A.5.2).

3. We then show that in autoregressive training, the expected number of times a token $t^*$ appears as context can differ from the expected number of times it is predicted, since these two quantities are affected differently by statistical correlations between tokens. In contrast, under bidirectional training, the expected counts are always equal, regardless of token correlations (Section A.5.3).

4. Finally, we combine the above results to prove our main theorems: (1) the weight updates to $\mathbf{W}_{qk}$ induce *column dominance* under autoregressive training (Section A.5.4-A.5.5), and (2) the weight updates to $\mathbf{W}_{qk}$ induce *symmetry* under bidirectional training (Section A.5.6).

A.5.1. PROOF OF PROPOSITION 2.2

First, we show that a token $t^*$ contributes differently to the updates of $\mathbf{W}_{qk}$ depending on whether it serves as context for predicting other tokens or is itself predicted.

*Proof.* Let $U = \{t_1, \ldots, t_N\}$ be a sequence of tokens with the embedding of every $i$-th token be given by $\mathbf{x}_i \in \mathbb{R}^d$. Proposition 2.1 shows that the implicit weight update can be decomposed with two equivalent regrouping of the double summations, as follows,

$$\Delta\mathbf{W}_{qk} = \sum_{(i,j)\in U} \beta_{ij}\mathbf{x}_i\mathbf{x}_j^T = \sum_j \left(\sum_{i\in\mathcal{P}_j} \beta_{ij}\mathbf{x}_i\right) \mathbf{x}_j^\top = \sum_i \mathbf{x}_i \left(\sum_{i\in\mathcal{C}_i} \beta_{ij}\mathbf{x}_j^\top\right), \tag{S40}$$

where $\mathcal{P}_i \subset U$ is the set of tokens predicted by a given token $t_i$, while $\mathcal{C}_j \subset U$ is the set of tokens that predict a given token $t_j$. We neglect any constant of proportionality - such as a learning rate - for simplicity, and we do not assume any specific structure on $\mathcal{P}_i$ and $\mathcal{C}_j$ (autoregressive training, bidirectional training, or others). First, the contribution of $t^* \in U$ to the weight update when $t^*$ is used as context to predict a set of tokens $\mathcal{P}_{t^*} \subset U$ is

$$\Delta\mathbf{W}_{qk}\Big|_{t_j=t^*} = \left(\sum_{i\in\mathcal{P}_{t^*}} \beta_{i*}\mathbf{x}_i\right) \mathbf{x}_{t^*}^\top. \tag{S41}$$

The associated weight update of the $k$-th column is then given by

$$\Delta\mathbf{w}_{\cdot,k}\Big|_{t_j=t^*} = \left(\sum_{i\in\mathcal{P}_{t^*}} \beta_{i*}\mathbf{x}_i\right) [\mathbf{x}_{t^*}]_k = [\mathbf{x}_{t^*}]_k \left(\sum_{i\in\mathcal{P}_{t^*}} \beta_{i*}\mathbf{x}_i\right), \tag{S42}$$

while the update of the $m$-th row is

$$\Delta\mathbf{w}_{m,\cdot}\Big|_{t_j=t^*} = \left(\sum_{i\in\mathcal{P}_{t^*}} \beta_{i*}[\mathbf{x}_i]_m\right) \mathbf{x}_{t^*}. \tag{S43}$$

A complementary argument can be made when predicting a given token $t^*$ from a set of tokens $\mathcal{C}_{t^*}$. Indeed, the contribution to the weight update when $t^*$ is predicted by set of tokens $\mathcal{C}_{t^*} \subset U$ is given by

$$\Delta\mathbf{W}_{qk}\Big|_{t_i=t^*} = \left(\sum_{j\in\mathcal{C}_{t^*}} \beta_{*j}\mathbf{x}_{t^*}\right) \mathbf{x}_j^\top. \tag{S44}$$

The associated weight update of the $k$-th column is then given by

$$\Delta\mathbf{w}_{k,\cdot}\Big|_{t_i=t^*} = \left(\sum_{j\in\mathcal{C}_{t^*}} \beta_{*j}[\mathbf{x}_j]_k\right) \mathbf{x}_{t^*}, \tag{S45}$$

while the update of the $m$-th row is

$$\Delta \mathbf{w}_{m,\cdot}\Big|_{t_i=t^*} = \left(\sum_{j\in\mathcal{C}_{t^*}} \beta_{*j}\mathbf{x}_j^\top\right)[\mathbf{x}_{t^*}]_m = [\mathbf{x}_{t^*}]_m\left(\sum_{j\in\mathcal{C}_{t^*}}\beta_{*j}\mathbf{x}_j^\top\right). \tag{S46}$$

Therefore, the weight update of each column (row) when a set of tokens predicts $t^*$ is equivalent to the weight update of each row (column) when $t^*$ is used to predict a set of tokens. This concludes the proof. $\qquad\square$

### A.5.2. ASYMMETRIC GROWTH OF ROWS AND COLUMNS DURING WEIGHT UPDATE

Next, we demonstrate that under reasonable assumptions about the statistical distribution of token embeddings, the expected norm of column updates exceeds that of row updates when the token $t^*$ is used as context to predict other tokens. Conversely, when $t^*$ is being predicted by other tokens, the row updates become dominant. To do so, we make the following assumptions:

- The token embeddings $\mathbf{x}_i$ are independent and identically distributed (i.i.d.) random vectors drawn from a probability distribution $\mathcal{D}$ with zero mean, $\mathbb{E}[\mathbf{x}_i] = \mathbf{0}$, and covariance matrix $\mathrm{Cov}(\mathbf{x}_i) = \Sigma$. This assumption holds at initialization for any Transformer model with learnable embeddings.
- The covariance matrix is not isotropic, i.e., $\Sigma \neq \sigma^2\mathbb{I},$. More specifically, we posit that there is partial alignment between the embeddings $\mathbf{x}_i$ due to the semantic and predictive relationships between tokens, which typically emerge during training.

Similar to Proposition 2.2, the scenarios where $t^*$ is used to predict other tokens and where $t^*$ is being predicted by other tokens are complementary. In the following Proposition, we focus solely on the case where $t^*$ serves as context to predict a set of tokens. A formal derivation for the opposite case where $t^*$ is predicted by other tokens is provided in a subsequent Corollary.

**Proposition A.7.** *(Asymmetric growth of columns and rows for context). Let $U = \{t_1,\ldots,t_N\}$ a sequence of tokens, and let $t^*$ be a token representing the context of every token $t_i \in U$. Let $\{\mathbf{x}_1\ldots,\mathbf{x}_N\}$ be the token embedding associated with $U$ such that each $\mathbf{x}_i \sim \mathcal{D}$ is a i.i.d. random vector drawn from a probability distribution $\mathcal{D}$ with zero mean and non-isotropic covariance $Cov(\mathbf{x}_i) = \Sigma$. Let $\Delta\mathbf{W}_{qk}$ be the weight update from Proposition 2.1. Then the squared norm of the $m$-th row and $k$-th column of $\mathbf{W}_{qk}$ satisfies*

$$\frac{\mathbb{E}_{\mathcal{D}}\left[||\Delta\mathbf{w}_{\cdot,k}||^2\right]}{\mathbb{E}_{\mathcal{D}}\left[||\Delta\mathbf{w}_{m,\cdot}||^2\right]} > 1 \quad \forall k \in \{1,\ldots,d\}, \forall m \text{ s.t. } \Sigma_{m,m} < \frac{Tr(\Sigma)}{d} \tag{S47}$$

*Proof.* Let $U = [t_0,\ldots,t_N]$ be a sequence of tokens, and let $t^*$ be the context of every token $t_i \in U$. Let $\{\mathbf{x}_i\}$ be the set of token embeddings associated with $U$ and let $\mathbf{y}$ be the token embedding of $t^*$. It follows from Proposition 2.2 that the squared norm of the weight update of the $k$-th column is given by

$$\Delta\mathbf{w}_{\cdot,k} = \sum_{i=1}^N \beta_{i*}\mathbf{x}_i[\mathbf{y}]_k \;\Rightarrow\; ||\Delta\mathbf{w}_{\cdot,k}||^2 = [\mathbf{y}]_k^2||\sum_{i=1}^N\beta_{i*}\mathbf{x}_i||^2, \tag{S48}$$

while the squared norm of the weight update of the $m$-th row by

$$\Delta\mathbf{w}_{m,\cdot} = \sum_{i=1}^N \beta_{i*}[\mathbf{x}_i]_m\mathbf{y} \;\Rightarrow\; ||\Delta\mathbf{w}_{m,\cdot}||^2 = \Big(\sum_{i=1}^N\beta_{i*}[\mathbf{x}_i]_m\Big)^2||\mathbf{y}||^2. \tag{S49}$$

Let $\{\mathbf{x}_i\} \sim \mathcal{D}$ be a set of i.i.d. random vectors from a probability distribution $\mathcal{D}$ such that $\mathbb{E}_{\mathcal{D}}[\mathbf{x}_i] = 0$ and $\mathrm{Cov}(\mathbf{x}_i) = \Sigma \in \mathbb{R}^{d,d}$. Therefore, each $k$-th coordinate $[\mathbf{x}_i]_k$ is such that $\mathbb{E}_{\mathcal{D}}[[\mathbf{x}_i]_k] = 0$ and $\mathrm{Var}([\mathbf{x}_i]_k) = \Sigma_{k,k}$. We also assume that $\mathbf{y}$ is statistically independent from $\mathbf{x}_i$ $\forall i$ with $\mathbb{E}[\mathbf{y}] = 0$ and covariance $\mathrm{Cov}(\mathbf{y}) = \sigma_y^2\mathbb{I}$. It follows that the expected value of $||\Delta\mathbf{w}_{\cdot,k}||^2$ over $\mathcal{D}$ is given by

$$\mathbb{E}_{\mathcal{D}}\left[||\Delta\mathbf{w}_{\cdot,k}||^2\right] = \mathbb{E}_{\mathcal{D}}\left[[\mathbf{y}]_k^2||\sum_{i=1}^N\beta_{i*}\mathbf{x}_i||^2\right] = \mathbb{E}_{\mathcal{D}}\left[[\mathbf{y}]_k^2\right]\mathbb{E}_{\mathcal{D}}\left[||\sum_{i=1}^N\beta_{i*}\mathbf{x}_i||^2\right]. \tag{S50}$$

Given the statistical independence between the entries of $\mathbf{x}_i$, the second term is equal to

$$
\begin{aligned}
\mathbb{E}_{\mathcal{D}}\left[||\sum_{i=1}^{N}\beta_{i*}\mathbf{x}_i||^2\right] &= \sum_{i,i'}\mathbb{E}_{\mathcal{D}}\left[\beta_{i*}\beta_{i'*}\mathbf{x}_i^{\top}\mathbf{x}_{i'}\right] \\
&= \sum_{i=1}^{N}\beta_{i*}^2\mathbb{E}_{\mathcal{D}}\left[\mathbf{x}_i^{\top}\mathbf{x}_i\right] + \sum_{i'\neq i}\beta_{i*}\beta_{i'*}\mathbb{E}_{\mathcal{D}}\left[\mathbf{x}_i^{\top}\mathbf{x}_{i'}\right] \\
&= \sum_{i=1}^{N}\beta_{i*}^2\mathrm{Tr}\left(\mathbb{E}_{\mathcal{D}}\left[\mathbf{x}_i\mathbf{x}_i^{\top}\right]\right) + \sum_{i'\neq i}\beta_{i*}\beta_{i'*}\mathbb{E}_{\mathcal{D}}\left[\mathbf{x}_i^{\top}\mathbf{x}_{i'}\right] \\
&= \mathrm{Tr}(\Sigma)\sum_{i=1}^{N}\beta_{i*}^2 \,,
\end{aligned}
\tag{S51}
$$

and therefore

$$
\mathbb{E}_{\mathcal{D}}\left[||\Delta\mathbf{w}_{\cdot,k}||^2\right] = \Gamma_{k,k}\mathrm{Tr}(\Sigma)\sum_{i=1}^{N}\beta_{i*}^2 \,.
\tag{S52}
$$

Similarly, the expected value of $||\Delta\mathbf{w}_{m,\cdot}||^2$ is given by

$$
\mathbb{E}_{\mathcal{D}}\left[||\Delta\mathbf{w}_{m,\cdot}||^2\right] = \mathbb{E}_{\mathcal{D}}\left[\left(\sum_{i=1}^{N}\beta_{i*}[\mathbf{x}_i]_m\right)^2||\mathbf{y}||^2\right] = \mathbb{E}_{\mathcal{D}}\left[\left(\sum_{i=1}^{N}\beta_{i*}[\mathbf{x}_i]_m\right)^2\right]\mathbb{E}_{\mathcal{D}}\left[||\mathbf{y}||^2\right] \,.
\tag{S53}
$$

The first term can be decomposed as

$$
\mathbb{E}_{\mathcal{D}}\left[\left(\sum_{i=1}^{N}\beta_{i*}[\mathbf{x}_i]_m\right)^2\right] = \sum_{i=1}^{N}\beta_{i*}^2\mathbb{E}_{\mathcal{D}}\left[[\mathbf{x}_i]_m^2\right] + \sum_{i'\neq i}\beta_{i*}\beta_{i'*}\mathbb{E}_{\mathcal{D}}\left[[\mathbf{x}_{i'}]_m[\mathbf{x}_i]_m\right] = \Sigma_{m,m}\sum_{i=1}^{N}\beta_{i*}^2 \,,
\tag{S54}
$$

and therefore

$$
\mathbb{E}_{\mathcal{D}}\left[||\Delta\mathbf{w}_{m,\cdot}||^2\right] = \Sigma_{m,m}\mathrm{Tr}(\Gamma)\sum_{i=1}^{N}\beta_{i*}^2 \,.
\tag{S55}
$$

The ratio of the expected value of these squared norms is

$$
\frac{\mathbb{E}_{\mathcal{D}}\left[||\Delta\mathbf{w}_{\cdot,k}||^2\right]}{\mathbb{E}_{\mathcal{D}}\left[||\Delta\mathbf{w}_{m,\cdot}||^2\right]} = \frac{\Gamma_{k,k}\mathrm{Tr}(\Sigma)}{\Sigma_{m,m}\mathrm{Tr}(\Gamma)} = \frac{\Gamma_{k,k}}{\mathrm{Tr}(\Gamma)}\frac{\mathrm{Tr}(\Sigma)}{\Sigma_{m,m}} = \frac{1}{d}\frac{\mathrm{Tr}(\Sigma)}{\Sigma_{m,m}} \,.
\tag{S56}
$$

We assume a non-isotropic covariance structure in $\Sigma$, that is, the average variance per dimension is lower than the total variance across all dimensions. This implies $\mathrm{Tr}(\Sigma) > d\,\Sigma_{m,m}$ for some $m$. It follows that,

$$
\frac{\mathbb{E}_{\mathcal{D}}\left[||\Delta\mathbf{w}_{\cdot,k}||^2\right]}{\mathbb{E}_{\mathcal{D}}\left[||\Delta\mathbf{w}_{m,\cdot}||^2\right]} = \frac{1}{d}\frac{\mathrm{Tr}(\Sigma)}{\Sigma_{m,m}} > \frac{1}{d}\frac{d\,\Sigma_{m,m}}{\Sigma_{m,m}} = 1 \quad \forall m \text{ s.t. } \Sigma_{m,m} < \frac{\mathrm{Tr}(\Sigma)}{d} \,,
\tag{S57}
$$

thus concluding the proof. $\square$

Again, a complementary argument can be made about the asymmetric weight update when a set of tokens $U = \{t_1, \ldots, t_N\}$ is used to predict a given token $t^*$. We formalize this in the following Corollary.

**Corollary A.8.** *(Asymmetric growth of columns and rows for prediction). Let $U = \{t_1, \ldots, t_N\}$ be a sequence of tokens, and let every token $t_i \in U$ be the context of a given token $t^*$. Let the associated embedding be drawn i.i.d. from a probability distribution $\mathcal{D}$ as in Proposition A.7. Let $\Delta\mathbf{W}_{qk}$ be the weight update from Proposition 2.1. Then the squared norm of the $m$-th row and $k$-th column of $\mathbf{W}_{qk}$ satisfies*

$$
\frac{\mathbb{E}_{\mathcal{D}}\left[||\Delta\mathbf{w}_{\cdot,k}||^2\right]}{\mathbb{E}_{\mathcal{D}}\left[||\Delta\mathbf{w}_{m,\cdot}||^2\right]} < 1 \quad \forall m \in \{1, \ldots, d\}, \forall k \text{ s.t. } \Sigma_{k,k} < \frac{\mathrm{Tr}(\Sigma)}{d}
\tag{S58}
$$

*Proof.* Let $U = [t_0, \ldots, t_N]$ a sequence of tokens, and let $t^*$ be predicted by every token $t_i \in U$. Let $\{\mathbf{x}_i\}$ be the set of token embeddings associated with $U$ and let $\mathbf{y}$ be the token embedding of $t^*$. It follows from Proposition 2.2 that the squared norm of the weight update of the $k$-th column is given by

$$\Delta \mathbf{w}_{\cdot,k} = \sum_{j=1}^{N} \beta_{*j}[\mathbf{x}_j]_k \mathbf{y} \ \Rightarrow \ ||\Delta \mathbf{w}_{\cdot,k}||^2 = \Big( \sum_{i=1}^{N} \beta_{*j}[\mathbf{x}_j]_k \Big)^2 ||\mathbf{y}||^2 \,, \tag{S59}$$

while the weight update of the $m$-th row follows

$$\Delta \mathbf{w}_{m,\cdot} = \sum_{j=1}^{N} \beta_{*j} \mathbf{x}_j [\mathbf{y}]_m \ \Rightarrow \ ||\Delta \mathbf{w}_{m,\cdot}||^2 = [\mathbf{y}]_m^2 || \sum_{i=1}^{N} \beta_{*j} \mathbf{x}_j ||^2 \,. \tag{S60}$$

Therefore, following the same arguments as in Proposition A.7, the ratio of the expected value of the square norms is given by

$$\frac{\mathbb{E}_{\mathcal{D}} \left[ ||\Delta \mathbf{w}_{\cdot,k}||^2 \right]}{\mathbb{E}_{\mathcal{D}} \left[ ||\Delta \mathbf{w}_{m,\cdot}||^2 \right]} = \frac{\Sigma_{k,k} \mathrm{Tr}(\Gamma)}{\Gamma_{m,m} \mathrm{Tr}(\Sigma)} = d \frac{\Sigma_{k,k}}{\mathrm{Tr}(\Sigma)} < \frac{d \, \Sigma_{k,k}}{d \, \Sigma_{k,k}} = 1 \quad \forall k \text{ s.t. } \Sigma_{k,k} < \frac{\mathrm{Tr}(\Sigma)}{d} \,, \tag{S61}$$

thus concluding the proof. $\qquad \square$

### A.5.3. Expected contribution for context and prediction and related remarks

Additionally, we show that in autoregressive training, the expected number of tokens predicted by a given token $t^*$ can differ from the number of tokens that predict $t^*$, with this ratio influenced by token correlations, while in bidirectional training, the two quantities are always equal regardless of such correlations, yielding a ratio of 1. We formalize this in the following proposition and illustrate, in the subsequent remark, an example from autoregressive training where the expected ratio exceeds 1.

**Proposition A.9.** *(Ratio of expected counts during autoregressive and bidirectional training).* *Let $\mathcal{V} = \{t_0, \ldots, t_V\}$ be a set of tokens. Let $\mathcal{U}$ be the sample space of all possible sequences of $N$ tokens, and let the sequence $U = \{t_1, \ldots, t_N\} \in \mathcal{U}$ be a random variable with probability distribution $P(U)$ defined over $\mathcal{U}$. Let $\Pr[t_j = t^*]$ be the probability that the token at index $j$ in a sequence $U \in \mathcal{U}$ is given by $t^* \in \mathcal{V}$. Let $\mathbb{E}_P[\mu_c(t^*)]$ be the expected number of tokens that are predicted by a given token $t^*$, and $\mathbb{E}_P[\mu_p(t^*)]$ the expected number of tokens that predict a given token $t^*$. For autoregressive training, the ratio*

$$\frac{\mathbb{E}_P[\mu_c(t^*, U)]}{\mathbb{E}_P[\mu_p(t^*, U)]} = \frac{\sum_{k=1}^{N} (N - k) \Pr[t_k = t^*]}{\sum_{k=1}^{N} (k - 1) \Pr[t_k = t^*]} \,, \tag{S62}$$

*depends on $\Pr[t_k = t^*] \ \forall t^* \in \mathcal{V}, \forall k \in \{1, \ldots. N\}$, while for bidirectional training the same ratio is given by*

$$\frac{\mathbb{E}_P[\mu_c(t^*, U)]}{\mathbb{E}_P[\mu_p(t^*, U)]} = 1 \,. \tag{S63}$$

*Proof.* Let $\mathcal{V} = \{t_0, \ldots, t_V\}$ be a set of tokens. Let $\mathcal{U} = \mathcal{V}^N$ denote the sample space of all possible sequences of $N$ tokens from the vocabulary $\mathcal{V}$. Let $\mathcal{F} = 2^{\mathcal{U}}$ be the $\sigma$-algebra given by the power set of $\mathcal{U}$, and let $P$ be the probability distribution over $\mathcal{U}$ that defines the distribution of the random variable sequence $U$. This defines the probability space $(\mathcal{U}, \mathcal{F}, P)$. For each position $k \in \{1, \ldots, N\}$ we let the random variable $T_k$ be $T_k : \mathcal{U} \to \mathcal{V}$ such that $T_k(U) \equiv t_k$. Let $E = \{U \in \mathcal{U} : t_k = t^*\}$ the event of all sequences where the $k$-th token is a fixed token $t^*$. The indicator function $\mathbb{1}\{E\}(U)$ is a random variable defined as

$$\mathbb{1}\{E\}(U) = \mathbb{1}\{t_k = t^*\}(U) = \begin{cases} 1 \text{ if } t_k = t^* \\ 0 \text{ otherwise} \,, \end{cases} \tag{S64}$$

and as such its expected value over $P$ is

$$\mathbb{E}_P \left[ \mathbb{1}\{t_k = t^*\}(U) \right] = \sum_{U \in \mathcal{U}} \mathbb{1}\{t_k = t^*\}(U) \Pr[U] = \sum_{U \in \mathcal{U} : t_k = t^*} \Pr[U] = \Pr[t_k = t^*] \,. \tag{S65}$$

Let $\mu_c(t^*, U)$ be the random variable quantifying the number of tokens predicted by $t^*$, while $\mu_p(t^*, U)$ is the random variable quantifying the number of tokens predicted by $t^*$. We analyzed the case of autoregressive and bidirectional training separately.

During autoregressive training, each time the token $t^*$ appears at position $k$ it is used as context to predict $N - k$ tokens, and it is predicted by $k - 1$ tokens. It follows that $\mu_c(t^*, U)$ is given by

$$\mu_c(t^*, U) = \sum_{l=2}^{N} \sum_{k=1}^{l-1} \mathbb{1}\{t_k = t^*\} = \sum_{k=1}^{N}(N - k)\mathbb{1}\{t_k = t^*\}, \tag{S66}$$

while $\mu_p(t^*, U)$ is given by

$$\mu_p(t^*, U) = \sum_{l=1}^{N-1} \sum_{k=1}^{l-1} \mathbb{1}\{t_l = t^*\} = \sum_{k=1}^{N}(k - 1)\mathbb{1}\{t_k = t^*\}. \tag{S67}$$

Therefore, the expected value of $\mu_c(t^*, U)$ over $P$ is

$$\mathbb{E}_P[\mu_c(t^*, U)] = \mathbb{E}_P\left[\sum_{k=1}^{N}(N - k)\mathbb{1}\{t_k = t^*\}\right] = \sum_{k=1}^{N}(N - k)\Pr[t_k = t^*], \tag{S68}$$

the expected value of $\mu_p(t^*)$ is

$$\mathbb{E}_P[\mu_p(t^*, U)] = \mathbb{E}_P\left[\sum_{k=1}^{N}(k - 1)\mathbb{1}\{t_k = t^*\}\right] = \sum_{k=1}^{N}(k - 1)\Pr[t_k = t^*], \tag{S69}$$

and their ratio is given by

$$\frac{\mathbb{E}_P[\mu_c(t^*, U)]}{\mathbb{E}_P[\mu_p(t^*, U)]} = \frac{\sum_{k=1}^{N}(N - k)\Pr[t_k = t^*]}{\sum_{k=1}^{N}(k - 1)\Pr[t_k = t^*]}. \tag{S70}$$

During bidirectional training, each time the token $t^*$ is masked at position $k$, it is predicted by $N - 1$ tokens. We assume that the token $t^*$ significantly contributes to the context of a masked token only if it is itself not masked (see Appendix A.3). Additionally, we assume that the probability $\rho$ of masking any given position $k$ is the same for all positions and does not depend on the specific token $t_k$ at that position. As such, we define masking as an independent Bernoulli variable $m_k : \Sigma_k \to \{0.1\} \sim P_m$ such that $\Pr[m_k = 1] = \rho$, independently of the token and sequence spaces. It follows that $\mu_c(t^*, U, m)$ is given by

$$\mu_c(t^*, U, m) = \sum_{k=1}^{N} \mathbb{1}\{t_k = t^*\}\mathbb{1}\{m_k = 0\} \sum_{k' \neq k} \mathbb{1}\{m_{k'} = 1\}, \tag{S71}$$

while $\mu_p(t^*, U, m)$ is given by

$$\mu_p(t^*, U, m) = \sum_{k=1}^{N} \mathbb{1}\{t_k = t^*\}\mathbb{1}\{m_k = 1\} \sum_{k' \neq k} \mathbb{1}\{m_{k'} = 0\}. \tag{S72}$$

Therefore, by taking the expectation with respect to both distributions $P$ and $P_m$, their respective expected values are given by

$$
\begin{aligned}
\mathbb{E}_{P,P_m}[\mu_c(t^*, U, m)] &= \mathbb{E}_{P,P_m}\left[\sum_{k=1}^{N} \mathbb{1}\{t_k = t^*\}\mathbb{1}\{m_k = 0\} \sum_{k' \neq k} \mathbb{1}\{m_{k'} = 1\}\right] \\
&= \sum_{k=1}^{N} \mathbb{E}_{P,P_m}\left[\mathbb{1}\{t_k = t^*\}\mathbb{1}\{m_k = 0\} \sum_{k' \neq k} \mathbb{1}\{m_{k'} = 1\}\right] \\
&= \sum_{k=1}^{N} \mathbb{E}_{P,P_m}\left[\mathbb{1}\{t_k = t^*\}\mathbb{1}\{m_k = 0\}\right] \sum_{k' \neq k}\left[\mathbb{1}\{m_{k'} = 1\}\right] \\
&= \sum_{k=1}^{N} \Pr(t_k = t^*)\Pr(m_k = 0) \sum_{k' \neq k} \Pr(m_{k'} = 1) \\
&= N\rho(N - 1)(1 - \rho) \sum_{k=1}^{N} \Pr(t_k = t^*),
\end{aligned}
\tag{S73}
$$

and

$$\mathbb{E}_{P,P_m}[\mu_p(t^*, U, m)] = \mathbb{E}_{P,P_m}\left[\sum_{k=1}^{N} \mathbb{1}\{t_k = t^*\}\mathbb{1}\{m_k = 1\}\sum_{k' \neq k}\mathbb{1}\{m_{k'} = 0\}\right]$$

(S74)

$$= N\rho(N-1)(1-\rho)\sum_{k=1}^{N}\Pr(t_k = t^*),$$

and their ratio is

$$\frac{\mathbb{E}_P[\mu_c(t^*, U)]}{\mathbb{E}_P[\mu_p(t^*, U)]} = 1,$$

(S75)

where we omit the dependence on the independent random variable $m$ for simplicity, thus concluding the proof. $\square$

The following remark highlights how, during autoregressive training, the ratio of expected token counts can vary depending on the correlations and conditional dependencies among the tokens.

*Remark* A.10. We separately analyze the following two cases:

- *No statistical correlation between tokens.* Assume that the tokens are statistically independent and identically distributed (i.i.d.) across positions, i.e., the probability of each token is the same at every position and does not depend on surrounding tokens. Under this assumption, during autoregressive training, the expected number of times token $t^*$ is used to predict future tokens is given by,

$$\mathbb{E}_P[\mu_c(t^*, U)] = \sum_{k=1}^{N}(N-k)\Pr[t_k = t^*] = \Pr[t^*]\sum_{k=1}^{N}(N-k) = \Pr[t^*]\frac{N(N-1)}{2}.$$

(S76)

  Similarly, the expected number of times token $t^*$ is predicted by preceding tokens is,

$$\mathbb{E}_P[\mu_p(t^*, U)] = \sum_{k=1}^{N}(k-1)\Pr[t_k = t^*] = \Pr[t^*]\sum_{k=1}^{N}(k-1) = \Pr[t^*]\frac{N(N-1)}{2}.$$

(S77)

  Therefore, when tokens are independent and identically distributed, the expected number of tokens predicted *by* $t^*$ and the expected number of tokens used to predict $t^*$ are equal. In this case, there is no difference between autoregressive and bidirectional training.

- *Statistical correlation between tokens.* Let us assume that the probability distribution $P$ encodes statistical dependencies between tokens. Define $\mu(t^*)$ as the expected position index of token $t^*$,

$$\mu(t^*) = \frac{\sum_{k=1}^{N} k\Pr[t_k = t^*]}{\sum_{k=1}^{N}\Pr[t_k = t^*]}.$$

(S78)

  Using this, the ratio of expected counts under autoregressive training becomes,

$$\frac{\mathbb{E}_P[\mu_c(t^*, U)]}{\mathbb{E}_P[\mu_p(t^*, U)]} = \frac{N - \mu(t^*)}{\mu(t^*) - 1}.$$

(S79)

  This expression shows that if $t^*$ tends to appear earlier in the sequence, that is, if $\mu(t^*) < (N+1)/2$ (left of center), then the ratio is greater than 1. Conversely, if $t^*$ appears later in the sequence (right of center), the ratio is less than 1. More formally, suppose there exists $\delta > 0$ such that

$$\mu(t^*) \leq \frac{N+1}{2} - \delta.$$

(S80)

  Then the ratio can be rewritten as,

$$\frac{\mathbb{E}_P[\mu_c(t^*, U)]}{\mathbb{E}_P[\mu_p(t^*), U]} - 1 = \frac{N + 1 - 2\mu(t^*)}{\mu(t^*) - 1},$$

(S81)

  where the numerator satisfies $N + 1 - 2\mu(t^*) \geq 2\delta$, and the denominator is positive for any token that can appear at position 1 (i.e., $\mu(t^*) > 1$). Hence, the ratio is bounded below by,

$$\frac{\mathbb{E}_P[\mu_c(t^*, U)]}{\mathbb{E}_P[\mu_p(t^*, U)]} \geq 1 + \varepsilon(t^*), \quad \text{with} \quad \varepsilon(t^*) = \frac{2\delta}{\mu(t^*) - 1}.$$

(S82)

### A.5.4. TAIL PROBABILITIES OF DISTRIBUTIONS WITH EQUAL MEAN AND DIFFERENT VARIANCES

Before proving column dominance in the next Theorem, we first establish that a probability distribution with higher variance is more likely to produce large values. Specifically, when two distributions share the same mean but differ in variance, the one with higher variance has a greater probability of generating samples that exceed a given threshold. We formalize this in the following lemma.

**Lemma A.11.** *Let $a$ and $b$ be two probability distributions with the same mean $\mu$, but different variances such that $\sigma_a^2 > \sigma_b^2$. Then, for all values $z > \sqrt{\sigma_a \sigma_b} - \mu$, the probability that a random sample from distribution $a$ exceeds $z$ is greater than or equal to the probability that a sample from distribution $b$ exceeds $z$, that is,*

$$\Pr[X_a > z] \geq \Pr[X_b > z] \quad \text{for all } z > \sqrt{\sigma_a \sigma_b} - \mu. \tag{S83}$$

*Proof.* Our goal is to find a $z$ such that

$$\Pr[X_a > z] \geq \Pr[X_b > z] \tag{S84}$$

for which we need the following steps: A lower bound for $\Pr[X_a > z]$, an upper bound for $\Pr[X_b > z]$, and a value of $z$ that makes the upper bound lower than the lower bound.

We start by using Chebyshev's inequality, to derive an upper bound for $\Pr[X_b > z]$,

$$\Pr[X_b > z] = \Pr[X_b - \mu > q\sigma_b] \leq \frac{1}{q^2} = \left(\frac{\sigma_b}{z - \mu}\right)^2. \tag{S85}$$

Now compute a lower bound for $\Pr[X_b > z]$ through the second moment method (which is very similar to the previous one, but inverted),

$$\Pr[X_a > z] = \Pr[X_a - z > 0] \geq \frac{(\mathbb{E}[X_a - z])^2}{\mathbb{E}\left[(X_a - z)^2\right]} = \left(\frac{\mu - z}{\sigma_a}\right)^2 \tag{S86}$$

Thus, we now only need to find $z$ satisfying

$$\left(\frac{\sigma_b}{z - \mu}\right)^2 < \left(\frac{z - \mu}{\sigma_a}\right)^2 \tag{S87}$$

for the case $z > \mu$, the inequality is fulfilled when

$$z > \sqrt{\sigma_a \sigma_b} - \mu. \tag{S88}$$

$\square$

### A.5.5. PROOF OF THEOREM 2.3

Here, we show that the weight updates of $\mathbf{W}_{qk}$ induce column dominance during autoregressive training, leading to the emergence of directionality. To do so, we make the following assumptions:

- We adopt the same assumptions stated in Propositions A.7 and A.9.
- At initialization, the entries of $\mathbf{W}_{qk}$ are drawn from a probability distribution $\mathcal{P}$ with finite mean $\mu$ and variance $\sigma^2$. This is satisfied by any standard machine learning initialization scheme. This second assumption implies that each $k$-th column $\mathbf{w}_{\cdot,k}$ and $m$-th row $\mathbf{w}_{m,\cdot}$ have the same mean $\mu$ and variance $\sigma^2/\sqrt{n}$. Furthermore, the squared norm of both rows and columns has equal mean $n(\sigma^2 + \mu^2)$ and variance $n(2\sigma^4 + 4\mu^2)$.

*Proof.* Let $\{U_1, \ldots, U_D\}$ be a dataset of sequences of length $N$ drawn i.i.d. from a distribution $P$. Let $k_u$ denote the position of token $t^*$ in sequence $U_u$, and let superscript $u$ refer to tokens in the $u$-th sequence. Following Proposition 2.2 and A.7, we obtain

$$\Delta \mathbf{w}_{\cdot,k} = \sum_{u=1}^{D} \left[ [\mathbf{y}]_k \sum_{i=k_u+1}^{N} \beta_{i*}^u \mathbf{x}_i^u + \sum_{j=1}^{k_u-1} \beta_{*j}^u [\mathbf{x}_j^u]_k \mathbf{y} \right], \tag{S89}$$

while the squared norm of the weight update of the $m$-th row by

$$\Delta\mathbf{w}_{m,\cdot} = \sum_{u=1}^{D}\left[\sum_{i=k_u+1}^{N}\beta_{i*}^u[\mathbf{x}_i^u]_m\mathbf{y} + [\mathbf{y}]_m\sum_{j=1}^{k_u-1}\beta_{*j}^u\mathbf{x}_j^u\right],\tag{S90}$$

where in both equations the superscript $u$ indicates tokens belonging to the $u$-th sequence in the dataset, and $k_u$ indicates the position index of $t^*$. It follows again from Proposition A.7 that the expected value of the squared norm of these updates is

$$\frac{\mathbb{E}_{\mathcal{D}}\left[||\Delta\mathbf{w}_{\cdot,k}||^2\right]}{\mathbb{E}_{\mathcal{D}}\left[||\Delta\mathbf{w}_{m,\cdot}||^2\right]} \approx \frac{\Gamma_{kk}\text{Tr}(\Sigma)\left(\sum_{u=1}^{D}\sum_{i=k_u+1}^{N}\beta_{i*}^u\right) + \Sigma_{kk}\text{Tr}(\Gamma)\left(\sum_{u=1}^{D}\sum_{j=1}^{k_u-1}\beta_{*j}^u\right)}{\Sigma_{mm}\text{Tr}(\Gamma)\left(\sum_{u=1}^{D}\sum_{i=k_u+1}^{N}\beta_{i*}^u\right) + \Gamma_{mm}\text{Tr}(\Sigma)\left(\sum_{u=1}^{D}\sum_{j=1}^{k_u-1}\beta_{*j}^u\right)}.\tag{S91}$$

Following Proposition A.9, let $\mu_c(t^*, U)$ be the random variable quantifying the number of tokens predicted by $t^*$, while $\mu_p(t^*, U)$ is the random variable quantifying the number of tokens predicted by $t^*$. When assuming autoregressive training, it follows that the empirical average of $\mu_c(t^*, U)$ and $\mu_p(t^*, U)$ over the dataset $\{U_1, \ldots, U_D\}$ approaches,

$$\frac{\mathbb{E}_P[\mu_c(t^*, U)]}{\mathbb{E}_P[\mu_p(t^*, U)]} = \frac{\sum_{j=1}^{N}(N-j)\Pr[t_j = t^*]}{\sum_{j=1}^{N}(j-1)\Pr[t_j = t^*]}.\tag{S92}$$

Following Remark A.10, assume that the expected position $\mu(t^*)$ of $t^*$ satisfies $\mu(t^*) < (N+1)/2 - \delta$ for some $\delta > 0$, then

$$\frac{\mathbb{E}_P[\mu_c(t^*, U)]}{\mathbb{E}_P[\mu_p(t^*, U)]} > 1 + \epsilon(t^*),\tag{S93}$$

for some $\epsilon(t^*) > 0$. Hence, across the dataset, $t^*$ appears more often as a context token than as a predicted token. This implies $\mathbb{E}_P[k_u] < (N+1)/2$, so that the first terms in the numerator and denominator of Equation (S91) dominate, leading to,

$$\frac{\mathbb{E}\left[||\Delta\mathbf{w}_{\cdot,k}||^2\right]}{\mathbb{E}\left[||\Delta\mathbf{w}_{m,\cdot}||^2\right]} > 1,\tag{S94}$$

that is, the net increase in column norms exceeds that of row norms. Assume $\mathbf{W}_{qk}$ is initialized with i.i.d. entries from a distribution with mean $\mu$ and variance $\sigma^2$, then at the beginning of training $\text{Var}(||\mathbf{w}_{\cdot,k}||) = \text{Var}(||\mathbf{w}_{m,\cdot}||)$, while after training we have $\text{Var}(||\mathbf{w}_{\cdot,k}||) > \text{Var}(||\mathbf{w}_{m,\cdot}||) \ \forall k, m$. Applying Lemma A.11, it follows that for all $w > \gamma$, where $\gamma = \sqrt{\text{Var}(||\mathbf{w}_{\cdot,k}||) \cdot \text{Var}(||\mathbf{w}_{m,\cdot}||)} - \mu$,

$$\Pr[||\mathbf{w}_{\cdot,k}|| > w] > \Pr[||\mathbf{w}_{m,\cdot}|| > w] \ \ \forall w > \gamma,\tag{S95}$$

thus concluding the proof. □

### A.5.6. PROOF OF THEOREM 2.4 AND RELATED REMARKS

Finally, we prove that the weight updates of $\mathbf{W}_{qk}$ induce symmetry during bidirectional training. Importantly, the column dominance is present only during autoregressive training. Indeed, it follows from Proposition A.9 that, during bidirectional training, the net increase of the norm of the columns is equal to the net increase of the norm of the rows. We formalize this in the following Corollary.

**Corollary A.12.** *(Bidirectional training does not induce directionality) Let $\mathcal{V} = \{t_0, \ldots, t_V\}$ be a vocabulary of tokens, and let $\mathcal{U} \subset \mathcal{V}^N$ denote the sample space of all sequences of length $N$. Let $U = \{t_1, \ldots, t_N\} \in \mathcal{U}$ be a random variable with $U \sim P(U)$. Define $\Pr[t_j = t^*]$ as the marginal probability that the token at position $j$ equals $t^* \in \mathcal{V}$. Let $\{\mathbf{x}_1, \ldots, \mathbf{x}_N\}$ be the token embeddings corresponding to the elements of $U$, where each embedding $\mathbf{x}_i \sim \mathcal{D}$ is drawn i.i.d. from a distribution $\mathcal{D}$ with zero mean and covariance matrix $Cov(\mathbf{x}_i) = \Sigma$.. Let $\mathbf{W}_{qk}$ be query-key matrix of a self-attention mechanism, and let $\Delta\mathbf{W}_{qk}$ denote its gradient update as defined in Proposition 2.1, computed under an autoregressive objective as in Definition A.3. Then, the variance of the norm of the rows and the columns at the end of training is equal,*

$$Var\left(||\mathbf{w}_{m,\cdot}||\right) = Var\left(||\mathbf{w}_{\cdot,k}||\right) \ \ \forall k, m.\tag{S96}$$

*Proof.* Let $\{U_1, \ldots, U_D\}$ be a dataset of sequences of length $N$ drawn i.i.d. from a distribution $P$. Let $k_u$ denote the position of token $t^*$ in sequence $U_u$, and let superscript $u$ refer to tokens in the $u$-th sequence. Following Proposition 2.2 and A.7, we obtain

$$\Delta\mathbf{w}_{\cdot,k} = \sum_{u=1}^{D} \left[ [\mathbf{y}]_k \sum_{i=k_u+1}^{N} \beta_{i*}^u \mathbf{x}_i^u + \sum_{j=1}^{k_u-1} \beta_{*j}^u [\mathbf{x}_j^u]_k \mathbf{y} \right], \tag{S97}$$

while the squared norm of the weight update of the $m$-th row by

$$\Delta\mathbf{w}_{m,\cdot} = \sum_{u=1}^{D} \left[ \sum_{i=k_u+1}^{N} \beta_{i*}^u [\mathbf{x}_i^u]_m \mathbf{y} + [\mathbf{y}]_m \sum_{j=1}^{k_u-1} \beta_{*j}^u \mathbf{x}_j^u \right], \tag{S98}$$

where in both equations the superscript $u$ indicates tokens belonging to the $u$-th sequence in the dataset, and $k_u$ indicates the position index of $t^*$. It follows again from Proposition A.7 that the expected value of the squared norm of these updates is

$$\frac{\mathbb{E}_{\mathcal{D}}\left[||\Delta\mathbf{w}_{\cdot,k}||^2\right]}{\mathbb{E}_{\mathcal{D}}\left[||\Delta\mathbf{w}_{m,\cdot}||^2\right]} \approx \frac{\Gamma_{kk}\mathrm{Tr}(\Sigma)\left(\sum_{u=1}^{D}\sum_{i=k_u+1}^{N}\beta_{i*}^u\right) + \Sigma_{kk}\mathrm{Tr}(\Gamma)\left(\sum_{u=1}^{D}\sum_{j=1}^{k_u-1}\beta_{*j}^u\right)}{\Sigma_{mm}\mathrm{Tr}(\Gamma)\left(\sum_{u=1}^{D}\sum_{i=k_u+1}^{N}\beta_{i*}^u\right) + \Gamma_{mm}\mathrm{Tr}(\Sigma)\left(\sum_{u=1}^{D}\sum_{j=1}^{k_u-1}\beta_{*j}^u\right)} . \tag{S99}$$

Following Proposition A.9, let $\mu_c(t^*, U)$ be the random variable quantifying the number of tokens predicted by $t^*$, while $\mu_p(t^*, U)$ is the random variable quantifying the number of tokens predicted by $t^*$. When assuming bidirectional training training, it follows that the empirical average of $\mu_c(t^*, U)$ and $\mu_p(t^*, U)$ over the dataset $\{U_1, \ldots, U_D\}$ approaches,

$$\frac{\mathbb{E}_P[\mu_c(t^*, U)]}{\mathbb{E}_P[\mu_p(t^*, U)]} = 1 . \tag{S100}$$

From Proposition A.7, it follows that the net increase in the norms of the columns and rows of $\mathbf{W}_{qk}$ is the same. Assuming the same initialization conditions as in Theorem 2.3, it follows that by the end of training,

$$\mathrm{Var}(\|\mathbf{w}_{m,\cdot}\|) = \mathrm{Var}(\|\mathbf{w}_{\cdot,k}\|) \quad \forall\, k, m , \tag{S101}$$

which concludes the proof. $\qquad\square$

We now prove Theorem 2.4, showing that the bidirectional training objective induces approximate symmetry in the gradient contributions of each token pair $(i, j)$. Specifically, we assume that predicting token $t_i$ from $t_j$ is correlated with predicting $t_j$ from $t_i$, that is, the terms $\beta_{ij}$ and $\beta_{ji}$ from Proposition 2.1 are correlated, leading to similar contributions in both directions.

*Proof.* Let $U = \{t_1, \ldots, t_N\}$ be a sequence of tokens. It follows from Proposition 2.1 that the implciit weight update for $\mathbf{W}_{qk}$ following the gradient of $\mathcal{L}(\mathbf{U})$ w.r.t. $\mathbf{W}_{qk}$ is given by

$$\Delta\mathbf{W}_{qk} = \sum_{i=1}^{N}\sum_{j=1}^{N} \beta_{ij}\mathbf{K}_{ij} , \tag{S102}$$

where we neglect any constant of proportionality (e.g. learning rate) for simplicity. The double summation in Equation (S102) contains $N^2$ elements. Note that $\mathbf{K}_{ji} = \mathbf{K}_{ij}^\top$, so we can rewrite the double summation as follows,

$$\sum_{i=1}^{N}\sum_{j=1}^{N}\beta_{ij}\mathbf{K}_{ij} = \sum_{i=1}^{N}\beta_{ii}\mathbf{K}_{ii} + \sum_{\substack{i,j=1\\i<j}}^{N}(\beta_{ij}\mathbf{K}_{ij} + \beta_{ji}\mathbf{K}_{ji}) \tag{S103}$$

where the first term includes the diagonal terms, and the second includes the contributions of every pair $(i, j)$ with $i, j \in [0, \ldots, N]$. The second term can be written as,

$$\sum_{\substack{i,j=1\\i<j}}^{N}(\beta_{ij}\mathbf{K}_{ij} + \beta_{ji}\mathbf{K}_{ji}) = \sum_{\substack{i,j=1\\i<j}}^{N}(\beta_{ij}\mathbf{K}_{ij} + \beta_{ji}\mathbf{K}_{ij}^\top), \tag{S104}$$

and by decomposing $\mathbf{K}_{ij}$ in its symmetric and skew-symmetric parts, such that $\mathbf{K}_{ij} = \mathbf{S}_{ij} + \mathbf{N}_{ij}$, we obtain,

$$
\begin{aligned}
\sum_{\substack{i,j=1 \\ i<j}}^{N} (\beta_{ij}\mathbf{K}_{ij} + \beta_{ji}\mathbf{K}_{ij}^{\top}) &= \sum_{\substack{i,j=1 \\ i<j}}^{N} \left[ \beta_{ij}(\mathbf{S}_{ij} + \mathbf{N}_{ij}) + \beta_{ji}(\mathbf{S}_{ij} + \mathbf{N}_{ij})^{T} \right] \\
&= \sum_{\substack{i,j=1 \\ i<j}}^{N} \left[ (\beta_{ij} + \beta_{ji})\mathbf{S}_{ij} + (\beta_{ij} - \beta_{ji})\mathbf{N}_{ij} \right].
\end{aligned}
\tag{S105}
$$

Let $\beta_{ij}$ and $\beta_{ji}$ be such that $\text{sign}(\beta_{ij}) = \text{sign}(\beta_{ji})$ and $|\beta_{ij}| \approx |\beta_{ji}|$. By defining $\Delta\mathbf{W}_{qk}\big|_{\mathbf{t}_i \leftrightarrow \mathbf{t}_j} = \beta_{ij}\mathbf{K}_{ij} + \beta_{ji}\mathbf{K}_{ij}^{T}$, it follows that

$$
\Delta\mathbf{W}_{qk}\big|_{\mathbf{t}_i \leftrightarrow \mathbf{t}_j} \approx \sum_{\substack{i,j=1 \\ i<j}}^{N} \beta_{ij}\mathbf{S}_{ij} = \sum_{\substack{i,j=1 \\ i<j}}^{N} \beta_{ij}\mathbf{S}_{ij}^{\top} = \Delta\mathbf{W}_{qk}^{\top}\big|_{\mathbf{t}_i \leftrightarrow \mathbf{t}_j}
\tag{S106}
$$

thus concluding the proof. $\qquad\square$

Encoder-only models are typically not trained to predict every token in a sequence, but rather a random subset of tokens, and the model can attend to tokens bidirectionally. This is usually called Masked Language Modeling (MLM) (Devlin et al., 2019; Liu et al., 2019; Lan et al., 2020; Warner et al., 2024). Therefore, only a subset of terms in the double summation of Equation (S105) has the symmetric properties described above. We generalize the proof to this case in the following.

*Remark* A.13. Let $\mathcal{C}_i = [0, 1, \dots, N]$ and let the summation indexed by $i$ to run over a random subset of tokens $M \subset [0, 1, \dots, N]$. The weight update of $\mathbf{W}_{qk}$ is then given by

$$
\Delta\mathbf{W}_{qk}^{l} = \sum_{i \in M} \sum_{j=1}^{N} \beta_{ij}^{l}\mathbf{K}_{ij}^{l-1} .
\tag{S107}
$$

The double summation contains $N|M|$ elements, where $|M|$ is the cardinality of the subset $M$. We can rewrite the double summation as follows,

$$
\sum_{i \in M} \sum_{j=1}^{N} \beta_{ij}^{l}\mathbf{K}_{ij}^{l-1} = \sum_{i \in M} \beta_{ii}^{l}\mathbf{K}_{ii}^{l-1} + \sum_{\substack{i,j \in M \\ i<j}} (\beta_{ij}^{l}\mathbf{K}_{ij}^{l-1} + \beta_{ji}^{l}\mathbf{K}_{ji}^{l-1}) + \sum_{i \in M} \sum_{j \in \bar{M}} \beta_{ij}^{l}\mathbf{K}_{ij}^{l-1},
\tag{S108}
$$

where the first term includes the diagonal terms, the second includes the contributions of the pairs $(i, j)$ with $i, j \in M$, and the third includes the remaining terms with $\bar{M} = [1, \dots, N] \setminus M$. The second term can be written as,

$$
\sum_{\substack{i,j \in M \\ i<j}} (\beta_{ij}^{l}\mathbf{K}_{ij}^{l-1} + \beta_{ji}^{l}\mathbf{K}_{ji}^{l-1}) = \sum_{\substack{i,j \in M \\ i<j}} (\beta_{ij}^{l}\mathbf{K}_{ij}^{l-1} + \beta_{ji}^{l}\mathbf{K}_{ij}^{l-1^{\top}}),
\tag{S109}
$$

and by decomposing $\mathbf{K}_{ij}^{l-1}$ in its symmetric and skew-symmetric parts, such that $\mathbf{K}_{ij}^{l-1} = \mathbf{S}_{ij}^{l-1} + \mathbf{N}_{ij}^{l-1}$, we obtain,

$$
\begin{aligned}
\sum_{\substack{i,j \in M \\ i<j}} (\beta_{ij}^{l}\mathbf{K}_{ij}^{l-1} + \beta_{ji}^{l}\mathbf{K}_{ij}^{l-1^{\top}}) &= \sum_{\substack{i,j \in M \\ i<j}} \left[ \beta_{ij}^{l}(\mathbf{S}_{ij}^{l-1} + \mathbf{N}_{ij}^{l-1}) + \beta_{ji}^{l}(\mathbf{S}_{ij}^{l-1} + \mathbf{N}_{ij}^{l-1})^{T} \right] \\
&= \sum_{\substack{i,j \in M \\ i<j}} \left[ (\beta_{ij}^{l} + \beta_{ji}^{l})\mathbf{S}_{ij}^{l-1} + (\beta_{ij}^{l} - \beta_{ji}^{l})\mathbf{N}_{ij}^{l-1} \right],
\end{aligned}
\tag{S110}
$$

with a similar structure as in Equation (S105).

Let $|M| = pN$ with $0 < p < 1$ being the percentage of tokens to be predicted during bidirectional training. The total number of pairs in the second term of Equation (S108) is given by a binomial coefficient, thus the total number of elements in the summation is $pN(pN - 1)$. The total number of elements in the third term is instead the product $pN(N - pN)$.

Therefore, the percentage of symmetric weight updates from the second term over the total number of updates in the third term is given by

$$\frac{pN(pN-1)}{pN(N-pN)} \approx \frac{pN}{(N-pN)}, \tag{S111}$$

in the limit of large $N$. In practice, $p$ is set to be around 15%-30% (Devlin et al., 2019; Liu et al., 2019; Lan et al., 2020; Warner et al., 2024), leading to $\approx 25\%$ of symmetric weight updates on average.

### A.6. Properties of the symmetry score in Definition 3.1 and related proofs

The score $s$ we introduce in Section 3 indicates the *degree* of symmetry of a matrix $\mathbf{M}$ by quantifying the contribution to the Frobenious norm of its symmetric and skew-symmetric parts. In particular, $s$ equals 1 and -1 for a fully symmetric and skew-symmetric matrix. Accordingly, positive (negative) values of $s$ indicate the presence of symmetric (skew-symmetric) structures. Here, we provide a proof for these properties. First, we show that the Frobenious norm of any square matrix $\mathbf{M}$ can be decomposed in the sum of the Frobenious norm of its symmetric and skew-symmetric components, as in the following Lemma,

**Lemma A.14.** *For any square matrix $\mathbf{M} \in \mathbb{M}_n$ the following equivalence holds*

$$||\mathbf{M}||_F^2 = ||\mathbf{M}_s||_F^2 + ||\mathbf{M}_n||_F^2. \tag{S112}$$

*Proof.* The Frobenius norm of a matrix $\mathbf{M}$ can be defined as $||\mathbf{M}||_F = \sqrt{\mathrm{Tr}(\mathbf{M}\mathbf{M}^\top)}$, and as such we observe that for any square matrix $\mathbf{M}$ we get

$$||\mathbf{M}||_F = \sqrt{\mathrm{Tr}(\mathbf{M}\mathbf{M}^\top)} = \sqrt{\mathrm{Tr}\big((\mathbf{M}_s + \mathbf{M}_n)(\mathbf{M}_s + \mathbf{M}_n)^\top\big)} =$$
$$= \sqrt{\mathrm{Tr}(\mathbf{M}_s\mathbf{M}_s^\top) + \mathrm{Tr}(\mathbf{M}_s\mathbf{M}_n^\top) + \mathrm{Tr}(\mathbf{M}_n\mathbf{M}_s^\top) + \mathrm{Tr}(\mathbf{M}_n\mathbf{M}_n^\top)}. \tag{S113}$$

It follows from the cyclic property of the trace operator that the mixing terms cancel out as follows,

$$\mathrm{Tr}(\mathbf{M}_s\mathbf{M}_n^\top) + \mathrm{Tr}(\mathbf{M}_n\mathbf{M}_s^\top) = -\mathrm{Tr}(\mathbf{M}_s\mathbf{M}_n) + \mathrm{Tr}(\mathbf{M}_s\mathbf{M}_n) = 0, \tag{S114}$$

resulting in

$$||\mathbf{M}||_F = \sqrt{\mathrm{Tr}(\mathbf{M}_s\mathbf{M}_s^\top) + \mathrm{Tr}(\mathbf{M}_n\mathbf{M}_n^\top)}. \tag{S115}$$

Therefore, as both terms on the right-hand side are semi-positive definite, we conclude the proof as follows,

$$||\mathbf{M}||_F^2 = \mathrm{Tr}(\mathbf{M}_s\mathbf{M}_s^\top) + \mathrm{Tr}(\mathbf{M}_n\mathbf{M}_n^\top) = ||\mathbf{M}_s||_F^2 + ||\mathbf{M}_n||_F^2. \tag{S116}$$

$\square$

Next, we formulate the properties of the symmetry score as follows,

**Proposition A.15.** *The symmetry score $s$ quantifies the degree of symmetry or skew-symmetry of a given square matrix $\mathbf{M}$. In particular,*
*1) The symmetry score $s$ is a scalar value bounded in the range $[-1, 1]$.*
*2) A symmetry score $s = \pm 1$ indicates a fully symmetric or skew-symmetric matrix, respectively.*
*3) The symmetry score of a random matrix $\mathbf{M} \in \mathbb{M}_n$ with entries $\mathbf{M}_{ij} \sim p(0, \sigma)$ from a probability distribution with zero mean and finite variance tends to zero as $8/n$ in the limit $n \to \infty$.*

*Proof.* To prove the points (1) and (2), we first show that it follows from Lemma A.14 that the squared Frobenious norm of $\mathbf{M}_s$ and $\mathbf{M}_n$ are in an orthogonal relation

$$||\mathbf{M}||_F = \sqrt{||\mathbf{M}_s||_F^2 + ||\mathbf{M}_n||_F^2}. \tag{S117}$$

Therefore, for any given $\mathbf{M}$, the norms $||\mathbf{M}_s||_2^2$ and $||\mathbf{M}_n||_2^2$ are such that a higher value of the first leads to to a lower value of the second, and vice versa. In particular, it is straightforward to observe that $||\mathbf{M}_s||_2 = ||\mathbf{M}||_2$ and $||\mathbf{M}_n||_2 = 0$ if $\mathbf{M}$ is

symmetric. Next, we derive a decomposition of the squared Frobenious norm of the symmetric and skew-symmetric part of $\mathbf{M}$. From the definition of $\mathbf{M}_s$ we obtain that

$$
\begin{aligned}
||\mathbf{M}_s||_F^2 = \operatorname{Tr}(\mathbf{M}_s \mathbf{M}_s^\top) &= \frac{1}{4} \operatorname{Tr}\big[(\mathbf{M} + \mathbf{M}^\top)(\mathbf{M}^\top + \mathbf{M})\big] \\
&= \frac{1}{4}\big[\operatorname{Tr}(\mathbf{M}\mathbf{M}^\top) + \operatorname{Tr}(\mathbf{M}\mathbf{M}) + \operatorname{Tr}(\mathbf{M}^\top\mathbf{M}^\top) + \operatorname{Tr}(\mathbf{M}^\top\mathbf{M})\big] \\
&= \frac{1}{2}\big[\operatorname{Tr}(\mathbf{M}\mathbf{M}^\top) + \operatorname{Tr}(\mathbf{M}\mathbf{M})\big] \\
&= \frac{1}{2}\big[||\mathbf{M}||_F^2 + \operatorname{Tr}(\mathbf{M}\mathbf{M})\big] .
\end{aligned}
\tag{S118}
$$

Since the upper bound for $||\mathbf{M}_s||_F^2$ is given by $||\mathbf{M}||_F^2$, the second term on the left-hand side has an upper bound given by,

$$
\operatorname{Tr}(\mathbf{M}\mathbf{M}) \leq \frac{1}{2}||\mathbf{M}||_F^2 ,
\tag{S119}
$$

A complementary relation holds for the skew-symmetric component of $\mathbf{M}$,

$$
||\mathbf{M_n}||_F^2 = \frac{1}{4}\operatorname{Tr}\big[(\mathbf{M} - \mathbf{M}^\top)(\mathbf{M}^\top - \mathbf{M})\big] = \frac{1}{2}\big[||\mathbf{M}||_F^2 - \operatorname{Tr}(\mathbf{M}\mathbf{M})\big] ,
\tag{S120}
$$

which, following the same logic, defines a lower-bound for $\operatorname{Tr}(\mathbf{M}\mathbf{M})$ as follows,

$$
-\frac{1}{2}||\mathbf{M}||_F^2 \leq \operatorname{Tr}(\mathbf{M}\mathbf{M}) .
\tag{S121}
$$

Given Definition 3.1 we can write,

$$
s = \frac{||\mathbf{M}||_F^2 + \operatorname{Tr}(\mathbf{M}\mathbf{M}) - ||\mathbf{M}||_F^2 + \operatorname{Tr}(\mathbf{M}\mathbf{M})}{||\mathbf{M}||_F^2} = 2\frac{\operatorname{Tr}(\mathbf{M}\mathbf{M})}{||\mathbf{M}||_F^2}
\tag{S122}
$$

and by combining the bounds derived previously we obtain,

$$
-1 \leq s \leq 1
\tag{S123}
$$

with

$$
\begin{cases}
s = 1 & \text{if} \quad \mathbf{M} = \mathbf{M}^\top \\
s = -1 & \text{if} \quad \mathbf{M} = -\mathbf{M}^\top
\end{cases}
\tag{S124}
$$

To prove the point (3), let each entry $m_{ij} = [\mathbf{M}]_{ij}$ be an independent, identically distributed sample from a random distribution with mean zero and a finite variance $\sigma^2$. We compute the Frobenius norm of the symmetric and skew-symmetric parts as follows,

$$
\begin{aligned}
||\mathbf{M}_s||_F^2 &= \sum_{i \neq j}(\mathbf{M}_{ij} + \mathbf{M}_{ji})^2 + \sum_i (2\mathbf{M}_{ii})^2 \\
||\mathbf{M}_n||_F^2 &= \sum_{i \neq j}(\mathbf{M}_{ij} - \mathbf{M}_{ji})^2 .
\end{aligned}
\tag{S125}
$$

Here, the skew-symmetric part has a zero diagonal term (because of the subtraction), and the symmetric part has twice the diagonal of the original matrix $\mathbf{M}$ (because of the addition). Since the entries are independent, $\mathbf{M}_{ij}$ is independent of $\mathbf{M}_{ji}$ for all $j \neq i$, and thus we can treat the off-diagonal entries of the $\mathbf{M}_s$ and $\mathbf{M}_n$ terms as a sum and difference of two independent random samples having mean zero and the same variance. It follows that the resulting distribution has a mean zero and a variance of $2\sigma^2$ in both cases,

$$
\sum_{i \neq j}(\mathbf{M}_{ij} \pm \mathbf{M}_{ji})^2 = 2\sum_{i=1}^n \sum_{j=i+1}^n (\mathbf{M}_{ij} \pm \mathbf{M}_{ji})^2 \underset{n \to \infty}{\approx} n(n-1)\operatorname{Var}[\mathbf{M}_{ij} \pm \mathbf{M}_{ji}] = n(n-1)2\sigma^2 ,
\tag{S126}
$$

where the approximation is due to the central limit theorem. Applying a similar logic to the second term on the symmetric norm, each entry is the double of a random i.i.d. distribution with

$$\sum_{i=1}^{N} (2\mathbf{M}_{ii})^2 \underset{N \to \infty}{\approx} n \text{Var}[\mathbf{M}_{ij}] = n4\sigma^2 . \tag{S127}$$

Finally, we take the Frobenius norm of the random matrix itself and apply the same logic, where there are $n^2$ entries with a variance of $\sigma^2$,

$$\|\mathbf{M}\|_F^2 \underset{n \to \infty}{\approx} n^2 \sigma^2 . \tag{S128}$$

It follows that the symmetry score is given by

$$s = 2 \frac{\|\mathbf{M}_s\|_F^2 - \|\mathbf{M}_n\|_F^2}{\|\mathbf{M}\|_F^2} \underset{n \to \infty}{\approx} \frac{8\sigma^2 n}{\sigma^2 n^2} = \frac{8}{n} , \tag{S129}$$

where the symmetry score is zero in the limit $n \to \infty$ with convergence from the positive side. $\qquad\square$

## A.7. Properties of the directionality score in Definition 3.2 and related proofs

The score $d$ we introduce in Section 3 quantifies the directional bias of a square matrix $\mathbf{M}$ by comparing the total norm of the "outliers" rows and columns, that is, that are higher than $\gamma$ times the standard deviations of the norms. A directionality score $d$ of 1 indicates the presence of rows with high "outlier" norms and the absence of outliers in the distribution of the column norms. The opposite is true for a directionality score $d$ of -1. Accordingly, positive (negative) values of $d$ indicate the presence of row (column) dominance in the matrix. Here, we provide a proof for these properties.

**Proposition A.16.** *The symmetry score $d$ provides a quantitative measure of the degree of directional bias in a given square matrix $\mathbf{M}$.*
*1) The directionality score $d$ is a scalar value that lies within the range $[-1, 1]$.*
*2) For any given $\gamma > 0$, a directionality score $d = \pm 1$ indicates that vectors satisfying the condition defined by $\gamma$ are exclusively present in the rows or the column distribution, respectively.*
*3) The directionality score of a random matrix $\mathbf{M} \in \mathbb{M}_n$ with entries $\mathbf{M}_{ij} \sim p(0, \sigma)$ from a probability distribution with zero mean and a variance that scales as $O(n^{-1})$ tends to zero in the limit $n \to \infty$.*

*Proof.* To prove that the directionality score is bounded in the interval $[-1, 1]$, note that $\bar{c}_{\mathbf{M}}, \bar{r}_{\mathbf{M}} > 0$ simply because they are sums of norms. As both are positive,

$$|\bar{c}_{\mathbf{M}} - \bar{r}_{\mathbf{M}}| < \bar{c}_{\mathbf{M}} < \bar{c}_{\mathbf{M}} + \bar{r}_{\mathbf{M}} \tag{S130}$$

and thus

$$\frac{\bar{c}_{\mathbf{M}} - \bar{r}_{\mathbf{M}}}{\bar{c}_{\mathbf{M}} + \bar{r}_{\mathbf{M}}} < 1 \tag{S131}$$

and taking the negative sign for the absolute value,

$$-\frac{\bar{r}_{\mathbf{M}} - \bar{c}_{\mathbf{M}}}{\bar{c}_{\mathbf{M}} + \bar{r}_{\mathbf{M}}} > -1 \Rightarrow \frac{\bar{c}_{\mathbf{M}} - \bar{r}_{\mathbf{M}}}{\bar{c}_{\mathbf{M}} + \bar{r}_{\mathbf{M}}} > -1. \tag{S132}$$

In the extremes $d = \pm 1$, the numerator must be equal to the denominator in absolute value, implying that either $\bar{r}_{\mathbf{M}}$ or $\bar{c}_{\mathbf{M}}$ are zero and the other is positive. For completeness, we define the score as zero if both are zero.

Finally, we study the case of a random matrix. We start by noting that the values of $\bar{r}_{\mathbf{M}}, \bar{c}_{\mathbf{M}}$ are interchanged when we take the transpose, hence

$$\bar{r}_{\mathbf{M}} = \bar{c}_{\mathbf{M}^\top} \tag{S133}$$

Regardless of the scaling of the matrix and the value $\gamma$, the key property of a random matrix is that all entries are drawn from the same distribution. Hence,

$$\Pr[\mathbf{M}_{ij} = x] = \Pr[\mathbf{M}_{ji} = x] \Rightarrow \Pr[\mathbf{M} = \mathbf{X}] = \Pr[\mathbf{M} = \mathbf{X}^\top] \tag{S134}$$

| Configuration | BERT | BERT-Mini | BERT-Large |
|---|---|---|---|
| *Hidden Size* | 768 | 256 | 1024 |
| *Intermediate Size* | 3072 | 1024 | 4096 |
| *Number of Attention Heads* | 12 | 4 | 16 |
| *Number of Hidden Layers* | 12 | 4 | 24 |
| *Attention Dropout Probability* | 0.1 | 0.1 | 0.1 |
| *Hidden Activation Function* | gelu | gelu | gelu |
| *Hidden Dropout Probability* | 0.1 | 0.1 | 0.1 |
| *Layer Normalization Epsilon* | 1e-12 | 1e-12 | 1e-12 |
| *Max Position Embeddings* | 512 | 512 | 512 |
| *Position Embedding Type* | absolute | absolute | absolute |
| *Vocabulary Size* | 30522 | 30522 | 30522 |

*Table 1.* Configurations for BERT, BERT-Mini, and BERT-Large models.

for $x$ and $\mathbf{X}$ being any arbitrary value or matrix. As a consequence,

$$\Pr\left[\bar{c}_{\mathbf{M}} = x\right] = \Pr\left[\bar{c}_{\mathbf{M}^{\top}} = x\right] = \Pr\left[\bar{r}_{\mathbf{M}} = x\right] \tag{S135}$$

where the last equality comes from Eq. S133. The main point here is that the probability distribution of both rows and columns is the same. Pushing this forward, the expected value of $\bar{r}_{\mathbf{M}} - \bar{c}_{\mathbf{M}}$ is

$$\mathrm{E}\left[\bar{r}_{\mathbf{M}} - \bar{c}_{\mathbf{M}}\right] = \mathrm{E}\left[\bar{r}_{\mathbf{M}}\right] - \mathrm{E}\left[\bar{c}_{\mathbf{M}}\right] = \int \bar{c}_{\mathbf{M}}\Pr\left[\mathbf{M}\right] d\mathbf{M} - \int \bar{r}_{\mathbf{M}}\Pr\left[\mathbf{M}\right] d\mathbf{M} \tag{S136}$$

$$= \int \bar{r}_{\mathbf{M}}\Pr\left[\mathbf{M}^{\top}\right] d\mathbf{M}^{\top} - \int \bar{r}_{\mathbf{M}}\Pr\left[\mathbf{M}\right] d\mathbf{M} = 0 \tag{S137}$$

Furthermore, the expected value of $\bar{r}_{\mathbf{M}} + \bar{c}_{\mathbf{M}}$ is strictly positive, since both values are positive. Thus, their ratio, the directionality score of a random matrix, is zero.

Notice that to be thorough we must show that their variance is bounded scales down. Since weight initialization has been extensively studied, we will just make a general reference to it here. In machine learning, all weights are initialized with zero mean and variances that scale as $O(n^{-1})$. As $\mathbf{M}$ is a product of two matrices with such scaling, each entry would consist of the sum of $n$ random variables, where each one has a scaling of $O(n^{-2})$ since it is the product of two random variables with an $O(n^{-1})$ scaling. Thus, the entries of $\mathbf{M}$ also have a scaling of $O(n^{-1})$. Applying the mean value theorem gives us the desired result.

$\square$

# B. Experimental Details

We trained three BERT models (Devlin et al., 2019) to examine the evolution of the symmetry and directionality scores throughout the training process. Detailed information regarding the training procedure is provided below.

## B.1. Models

We train the standard BERT model (referred to as *BERT*), a smaller version (referred to as *BERT-Mini*), and a larger version (referred to as *BERT-Large*), following the implementation by (Devlin et al., 2019). Table 1 provides an overview of the model parameters. The standard BERT model has 12 layers, 12 attention heads, and embedding dimensions of 768 for the hidden layers and 3072 for the intermediate layers. In contrast, the smaller BERT-Mini model uses 4 layers, 4 attention heads, and embedding dimensions of 256 and 1024, respectively. The larger BERT-Large model features 24 layers, 16 attention heads, and embedding dimensions of 1024 and 4096 for the hidden and intermediate layers, respectively.

**Initialization** We optionally initialize the models using symmetric attention weights. Specifically, in each self-attention block, the query weight matrix $\mathbf{W_q}$ is initialized randomly, and the key weight matrix is set equal to it, $\mathbf{W_k} = \mathbf{W_q}$. This ensures that the key and query matrices are identical at initialization, introducing symmetry into the attention mechanism.

| Configuration | ViT (6 layers) CIFAR-10 | ViT (12 layers) ImageNet-1k |
|---|---|---|
| *Hidden Size* | 512 | 768 |
| *Intermediate Size* | 2048 | 3072 |
| *Number of Attention Heads* | 8 | 12 |
| *Number of Hidden Layers* | 6 | 12 |
| *Attention Dropout Probability* | 0.0 | 0.0 |
| *Hidden Activation Function* | gelu | gelu |
| *Hidden Dropout Probability* | 0.0 | 0.0 |
| *Layer Normalization Epsilon* | 1e-6 | 1e-6 |
| *Patch Size* | 4 | 16 |
| *QKV Bias* | true | true |
| *Encoder Stride* | 16 | 16 |

*Table 2.* Configurations for the 6-layer and 12-layer vision transformer models.

### B.2. Datasets

The models are trained on three datasets. First, we use the "20220301.en" snapshot from the Wikipedia dataset, which consists of 6.46 million samples crawled from Wikipedia. Second, we utilize the Jigsaw dataset with 159K samples, originally collected for a toxic comment classification challenge. Finally, we train on the English "2023-14" snapshot of the RedPajama-V2 dataset, which contains approximately 5.12 billion samples.

### B.3. Training Settings

The models are trained for $200,000$ update steps with a batch size of $32$ and $8$ gradient accumulation steps, effectively increasing the batch size to $256$ before each parameter update. The optimization is done using the AdamW optimizer (Loshchilov & Hutter, 2019), and the training schedule includes $200$ warmup steps to stabilize early training, followed by a linear decay learning rate schedule, starting at an initial learning rate of $5 \times 10^{-5}$ and weight decay of $0.01$. Mixed precision (fp16) training was utilized to maximize training efficiency, which reduces memory consumption and speeds up computation without significant loss of precision. The training data was processed with a masked language modeling (MLM) probability of 15%, ensuring that 15% of tokens were masked during training. The models are trained in the encoder and decoder mode, i.e., to predict masked tokens and subsequent tokens respectively.

## C. Enforcing symmetry at initialization for vision transformers

In parallel to the experiments described in Section 4.2, where symmetric initialization of the query-key weight matrix $\mathbf{W}_{qk}$ is investigated for BERT models, we extend the investigation to the vision domain. Specifically, we evaluate the impact of symmetric initialization in vision transformers (ViTs) (Dosovitskiy et al., 2021), training a 6-layer ViT on CIFAR-10 (Krizhevsky, 2009) and a 12-layer ViT on ImageNet-1k (Deng et al., 2009), using the same initialization strategy as applied to BERT models (see Appendix B.1 for implementation details). The architectural configurations for both models are summarized in Table 2.

Training is performed using the AdamW optimizer (Loshchilov & Hutter, 2019) with weight decay and a cosine annealing learning rate schedule. A linear warm-up phase is used during the first 30 epochs, with an initial learning rate scaled by a factor of $0.033$. The ImageNet-1k model is trained for 200 epochs using a per-device batch size of $256$ and gradient accumulation over two steps, while the CIFAR-10 model is trained for 500 epochs with a batch size of $128$ and no gradient accumulation. Both setups use a base learning rate of $0.003$, optimizer hyperparameters $\beta_1 = 0.9$ and $\beta_2 = 0.999$, and automatic mixed-precision training (fp16). We apply advanced data augmentation tailored to each dataset, including mixup ($\alpha = 0.2$) (Zhang et al., 2018), cutmix ($\alpha = 1.0$) (Yun et al., 2019), and label smoothing ($0.11$ for ImageNet-1k and $0.01$ for CIFAR-10). An exponential moving average (EMA) of the model weights (Tarvainen & Valpola, 2017) is maintained throughout training (decay rate = $0.99998$, update frequency = $32$ steps).

Contrary to the trends observed in our language model experiments, symmetric initialization does not result in faster

*Table 3.* Final training and evaluation metrics for ViT models on CIFAR-10 (Krizhevsky, 2009) and ImageNet-1k (Deng et al., 2009), trained with and without symmetric initialization. Evaluation metrics include final loss and top-1 accuracy. For each dataset, we compare models trained with standard initialization and with symmetric initialization.

| DATASET | TRAIN LOSS | EVAL LOSS | EVAL ACCURACY |
|---|---|---|---|
| **CIFAR-10** | | | |
| VIT 6-LAYERS | 1.07 | 0.40 | 0.90 |
| VIT 6-LAYERS (+ SYMM.) | 1.08 | 0.42 | 0.89 |
| **IMAGENET-1K** | | | |
| VIT 12-LAYERS | 2.49 | 1.97 | 0.76 |
| VIT 12-LAYERS (+ SYMM.) | 2.49 | 1.96 | 0.76 |

convergence or improved final performance for the vision transformer models. As shown in Table 3, training and evaluation losses, as well as top-1 accuracy, remained nearly identical between the standard and symmetric initialization conditions across both datasets. Importantly, the results with symmetric initialization are not degraded, indicating that such initialization is at least performance-neutral in our vision setting. While our findings suggest that symmetric initialization does not yield immediate benefits for ViTs under the given training regime, we do not rule out the possibility that alternative configurations or optimization strategies could leverage its potential. Further research may uncover conditions under which vision transformer architectures can benefit more substantially from symmetric initialization.

## D. Supplementary Figures

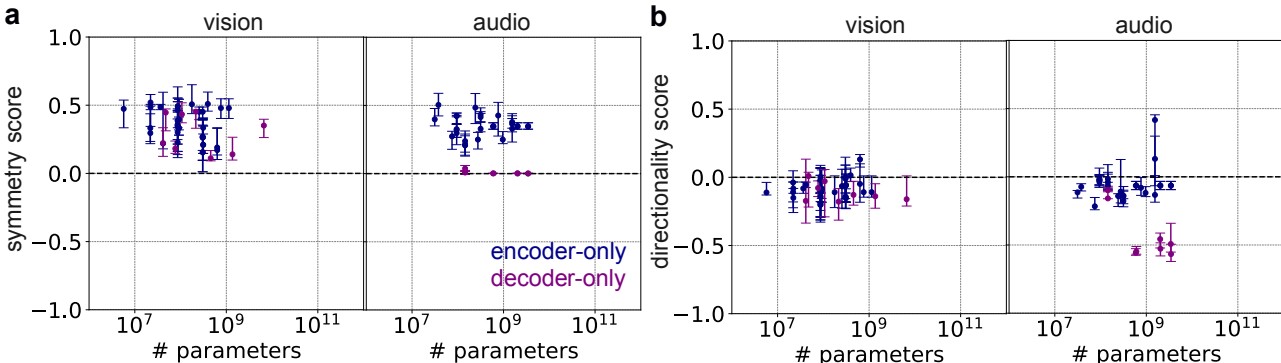

*Figure S1.* **a**) Median symmetry score of the matrix $\mathbf{W}_{qk}$ as a function of the total number of parameters for vision and audio models. Each dot corresponds to the median and the interquartile range across layers of a given pre-trained model. **b**) Same as in **a** for the median directionality score of the matrix $\mathbf{W}_{qk}$.

## E. Supplementary Tables

*Table 4.* Symmetry score for open-source pretrained language models. All models are available on Hugging Face (Wolf et al., 2020).

| Model | Median | Interquartile range | Model | Median | Interquartile range |
|---|---|---|---|---|---|
| BERT-tiny | 0.77 | ± [0.07, 0.07] | GPT-Neo1.3B | 0.14 | ± [0.03, 0.03] |
| BERT-mini | 0.62 | ± [0.03, 0.05] | GPT-Neo2.7B | 0.13 | ± [0.02, 0.04] |
| BERT-small | 0.69 | ± [0.10, 0.08] | GPT-J6B | 0.11 | ± [0.02, 0.03] |
| BERT-medium | 0.60 | ± [0.01, 0.02] | OpenAI-GPT | 0.07 | ± [0.04, 0.03] |
| BERT-base | 0.51 | ± [0.09, 0.07] | GPT2-XL | 0.12 | ± [0.03, 0.05] |
| BERT-large | 0.44 | ± [0.03, 0.08] | DistilGPT2 | 0.19 | ± [0.05, 0.05] |
| DistilBERT | 0.43 | ± [0.10, 0.13] | GPT-Neo125M | 0.14 | ± [0.09, 0.14] |
| BERT-2L-128 | 0.77 | ± [0.07, 0.07] | GPT-Neo1.3B | 0.14 | ± [0.03, 0.03] |
| BERT-4L-256 | 0.62 | ± [0.03, 0.05] | GPT-Neo2.7B | 0.14 | ± [0.03, 0.02] |
| BERT-4L-512 | 0.69 | ± [0.10, 0.08] | GPT-J6B | 0.11 | ± [0.02, 0.03] |
| BERT-8L-512 | 0.60 | ± [0.01, 0.02] | LLaMA2-7B | 0.12 | ± [0.02, 0.03] |
| BERT-base | 0.51 | ± [0.09, 0.07] | LLaMA2-13B | 0.17 | ± [0.02, 0.02] |
| BERT-large | 0.44 | ± [0.03, 0.08] | LLaMA3-8B | 0.00 | ± [0.00, 0.01] |
| DistilBERT | 0.43 | ± [0.10, 0.13] | LLaMA3.1-8B | 0.00 | ± [0.00, 0.01] |
| BEiT-base | 0.40 | ± [0.08, 0.02] | LLaMA3.2-8B | 0.01 | ± [0.01, 0.01] |
| BEiT-large | 0.33 | ± [0.05, 0.07] | LLaMA3.2-1B | 0.01 | ± [0.01, 0.01] |
| BEiT-base | 0.39 | ± [0.23, 0.07] | LLaMA3.2-3B | 0.01 | ± [0.01, 0.01] |
| BEiT-large | 0.26 | ± [0.17, 0.13] | LLaMA2-7B-chat | 0.12 | ± [0.02, 0.03] |
| BEiT-base | 0.39 | ± [0.23, 0.07] | LLaMA2-70B | 0.02 | ± [0.01, 0.02] |
| BEiT-large | 0.26 | ± [0.17, 0.13] | LLaMA2-7B-chat | 0.12 | ± [0.02, 0.03] |
| BEiT-large | 0.26 | ± [0.17, 0.13] | LLaMA2-13B-chat | 0.17 | ± [0.02, 0.02] |
| ALBERT-base | 0.72 | ± [0.00, 0.00] | LLaMA3-8B | 0.00 | ± [0.00, 0.00] |
| ALBERT-large | 0.70 | ± [0.00, 0.00] | LLaMA3-70B | 0.02 | ± [0.01, 0.01] |
| ALBERT-xlarge | 0.59 | ± [0.00, 0.00] | LLaMA3.1-8B | 0.00 | ± [0.00, 0.01] |
| ALBERT-xxlarge | 0.46 | ± [0.00, 0.00] | LLaMA3.1-70B | 0.01 | ± [0.00, 0.01] |
| RoBERTa-base | 0.49 | ± [0.03, 0.06] | LLaMA3.1-405B | 0.03 | ± [0.01, 0.03] |
| RoBERTa-large | 0.47 | ± [0.06, 0.06] | LLaMA3.2-1B | 0.00 | ± [0.00, 0.00] |
| XLM-R-base | 0.51 | ± [0.05, 0.03] | LLaMA3.2-3B | 0.01 | ± [0.01, 0.01] |
| XLM-R-large | 0.49 | ± [0.16, 0.12] | Mistral-7B | 0.00 | ± [0.00, 0.01] |
| RoBERTa-mnli | 0.47 | ± [0.06, 0.06] | Mixtral-8x22B | 0.00 | ± [0.00, 0.00] |
| DistilRoBERTa | 0.53 | ± [0.02, 0.06] | MobileLLM125M | 0.03 | ± [0.02, 0.03] |
| ModernBERT-base | 0.18 | ± [0.06, 0.18] | MobileLLM350M | 0.01 | ± [0.01, 0.01] |
| GPT1 | 0.07 | ± [0.04, 0.03] | Phi-1.5 | 0.09 | ± [0.03, 0.03] |
| GPT2 | 0.15 | ± [0.02, 0.03] | Phi-1 | 0.14 | ± [0.02, 0.01] |
| GPT2-medium | 0.17 | ± [0.03, 0.05] | Phi-2 | 0.07 | ± [0.03, 0.06] |

*Table 5.* Directionality score for open-source pretrained language models. All models are available on Hugging Face ([Wolf et al., 2020](#)).

| Model | Median | Interquartile range | Model | Median | Interquartile range |
|---|---|---|---|---|---|
| BERT-tiny | -0.79 | ± [0.11, 0.11] | GPT-Neo1.3B | -0.49 | ± [0.19, 0.13] |
| BERT-mini | -0.33 | ± [0.03, 0.04] | GPT-Neo2.7B | -0.57 | ± [0.15, 0.16] |
| BERT-small | -0.22 | ± [0.04, 0.03] | GPT-J6B | -0.28 | ± [0.09, 0.08] |
| BERT-medium | -0.23 | ± [0.06, 0.10] | OpenAI-GPT | -0.18 | ± [0.08, 0.07] |
| BERT-base | -0.08 | ± [0.02, 0.03] | GPT2-XL | -0.23 | ± [0.11, 0.10] |
| BERT-large | -0.03 | ± [0.02, 0.06] | DistilGPT2 | -0.51 | ± [0.03, 0.07] |
| DistilBERT | -0.13 | ± [0.00, 0.06] | GPT-Neo125M | -0.56 | ± [0.21, 0.08] |
| BERT-2L-128 | -0.79 | ± [0.11, 0.11] | GPT-Neo1.3B | -0.49 | ± [0.19, 0.13] |
| BERT-4L-256 | -0.33 | ± [0.03, 0.04] | GPT-Neo2.7B | -0.49 | ± [0.15, 0.21] |
| BERT-4L-512 | -0.22 | ± [0.04, 0.03] | GPT-J6B | -0.28 | ± [0.09, 0.08] |
| BERT-8L-512 | -0.23 | ± [0.06, 0.10] | LLaMA2-7B | -0.26 | ± [0.09, 0.13] |
| BERT-base | -0.08 | ± [0.02, 0.03] | LLaMA2-13B | -0.15 | ± [0.11, 0.03] |
| BERT-large | -0.03 | ± [0.02, 0.06] | LLaMA3-8B | -0.65 | ± [0.13, 0.20] |
| DistilBERT | -0.13 | ± [0.00, 0.06] | LLaMA3.1-8B | -0.64 | ± [0.17, 0.19] |
| BEiT-base | -0.10 | ± [0.06, 0.15] | LLaMA3.2-8B | -0.59 | ± [0.18, 0.22] |
| BEiT-large | -0.15 | ± [0.08, 0.07] | LLaMA3.2-1B | -0.59 | ± [0.18, 0.22] |
| BEiT-base | -0.14 | ± [0.15, 0.21] | LLaMA3.2-3B | -0.77 | ± [0.08, 0.19] |
| BEiT-large | -0.14 | ± [0.04, 0.14] | LLaMA2-7B-chat | -0.29 | ± [0.07, 0.14] |
| BEiT-base | -0.14 | ± [0.15, 0.21] | LLaMA2-70B | -0.24 | ± [0.10, 0.06] |
| BEiT-large | -0.14 | ± [0.04, 0.14] | LLaMA2-7B-chat | -0.29 | ± [0.07, 0.14] |
| BEiT-large | -0.15 | ± [0.03, 0.14] | LLaMA2-13B-chat | -0.19 | ± [0.12, 0.04] |
| ALBERT-base | -0.07 | ± [0.00, 0.00] | LLaMA3-8B | 0.01 | ± [0.05, 0.05] |
| ALBERT-large | -0.17 | ± [0.00, 0.00] | LLaMA3-70B | -0.37 | ± [0.09, 0.12] |
| ALBERT-xlarge | -0.24 | ± [0.00, 0.00] | LLaMA3.1-8B | -0.57 | ± [0.16, 0.13] |
| ALBERT-xxlarge | -0.15 | ± [0.00, 0.00] | LLaMA3.1-70B | -0.37 | ± [0.08, 0.11] |
| RoBERTa-base | -0.12 | ± [0.11, 0.03] | LLaMA3.1-405B | -0.17 | ± [0.07, 0.07] |
| RoBERTa-large | -0.06 | ± [0.03, 0.03] | LLaMA3.2-1B | -0.02 | ± [0.13, 0.08] |
| XLM-R-base | -0.02 | ± [0.02, 0.02] | LLaMA3.2-3B | -0.70 | ± [0.07, 0.22] |
| XLM-R-large | -0.02 | ± [0.03, 0.02] | Mistral-7B | -0.58 | ± [0.15, 0.13] |
| RoBERTa-mnli | -0.06 | ± [0.03, 0.03] | Mixtral-8x22B | -0.66 | ± [0.09, 0.16] |
| DistilRoBERTa | -0.14 | ± [0.09, 0.07] | MobileLLM125M | -0.13 | ± [0.15, 0.10] |
| ModernBERT-base | -0.04 | ± [0.05, 0.04] | MobileLLM350M | -0.34 | ± [0.13, 0.23] |
| GPT1 | -0.18 | ± [0.08, 0.07] | Phi-1.5 | -0.28 | ± [0.22, 0.19] |
| GPT2 | -0.58 | ± [0.06, 0.14] | Phi-1 | -0.40 | ± [0.03, 0.04] |

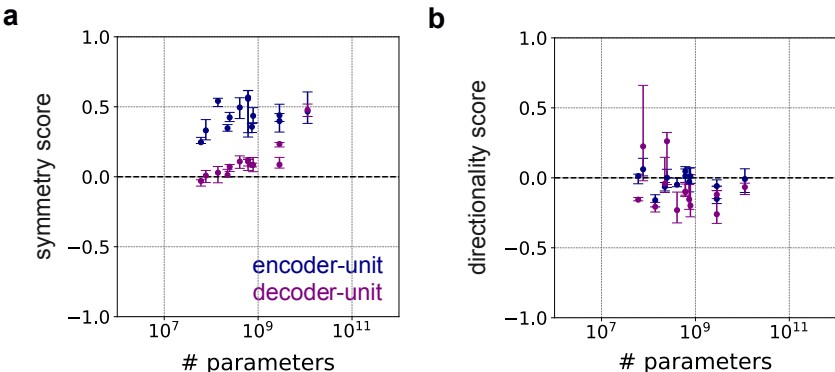

*Figure S2.* **a**) Median symmetry score of the matrix $\mathbf{W}_{qk}$ as a function of the total number of parameters for vision and audio models. Each dot corresponds to the median and the interquartile range across layers of the encoder component (blue) and decoder component (purple) of an encoder-decoder Transformer model. The encoder component of these models shows a high degree of symmetry compared to the decoder component. **b**) Same as in **a** for the median directionality score of the matrix $\mathbf{W}_{qk}$. The encoder and decoder components of these models do not show significant differences in directionality scores.

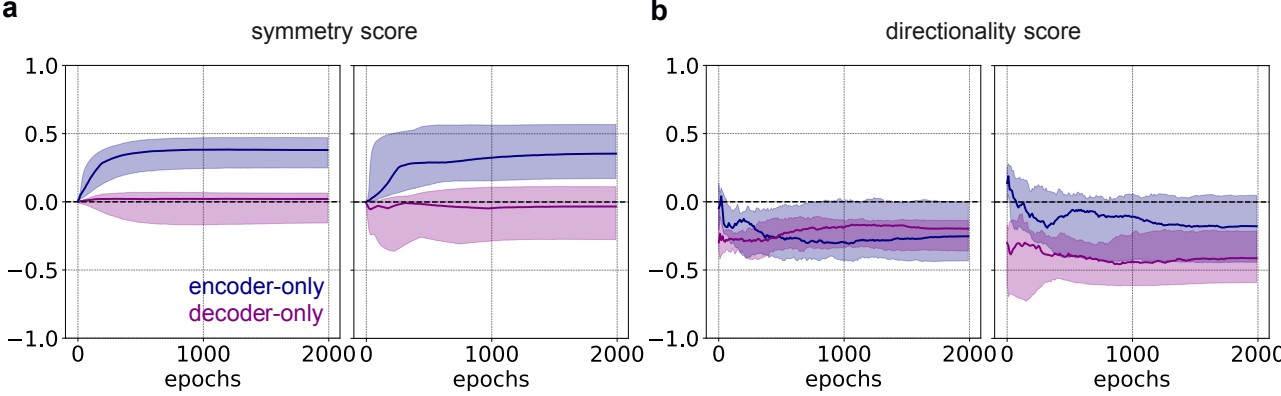

*Figure S3.* **a**) Evolution of symmetry score during training. Shown are the median and the interquartile range. Models were trained on the Jigsaw dataset (cjadams et al., 2017) (left) and on the Red Pajama dataset (Computer, 2023) (right). Encoder-only and decoder-only models are color-coded in blue and purple, respectively (see legend). **b**) Same as in panel **a** for the median directionality score.

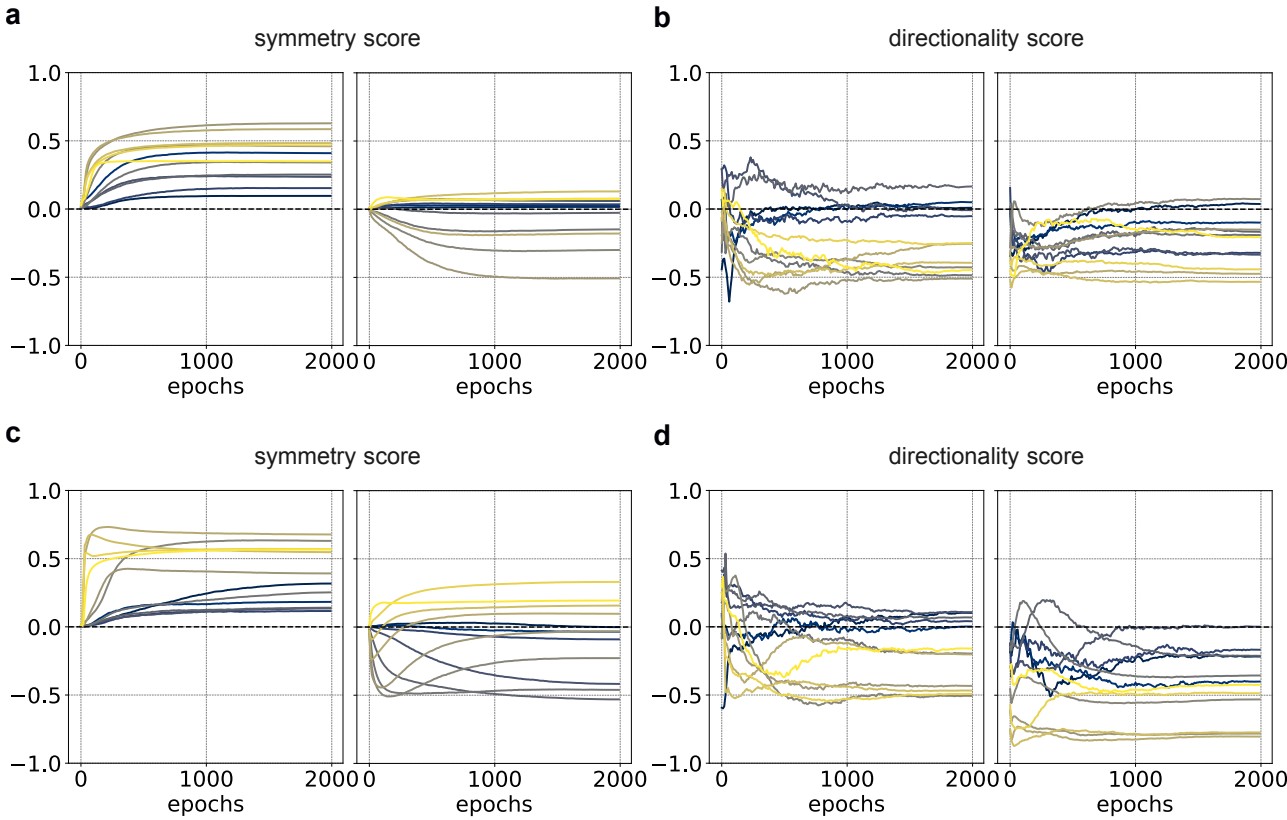

*Figure S4.* **a**) Evolution of the median symmetry score across layers of the encoder-only (left) and decoder-only (right) models. Each layer is color-coded as shown on the legend of Figure 3. Shown are the median and the interquartile range. Models were trained on the Jigsaw dataset (cjadams et al., 2017). **b**) Same as in panel **a** for the median directionality score. **c**) Same as in panel **a** for models trained on the Red Pajama dataset (Computer, 2023). **d**) Same as panel **c** for the median directionality score.

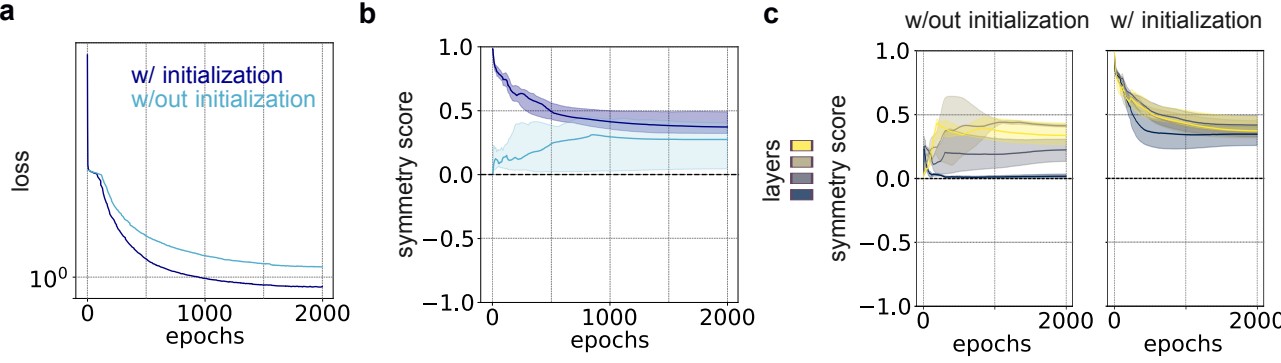

*Figure S5.* **a**) Loss a 4-layer encoder-only model with (dark blue) and without (light blue) symmetric initialization, respectively. Models are trained on the Red Pajama dataset (Computer, 2023). **b**) Median symmetry score during training Color code is as in panel **a**. **c**) Same as in panel **a** for each layer in the model with and without symmetric initialization. Each layer is color-coded as in the illustration on the left. All plots show the median and the interquartile range across the heads of a given layer.

