# OpenReview forum: "The underlying structures of self-attention: symmetry, directionality, and emergent dynamics in Transformer training"
_ICML.cc/2025/Conference — ICML 2025 poster_

### Official Review · Reviewer_ydUg · 2025-03-11

**Overall Recommendation:** 4

**Summary:**

The paper studies how bidirectional and autoregressive training objectives influence the structure of the query-key matrix $W_{qk}$ in self-attention. The results show that bidirectional training induces symmetric structures in $W_{qk}$, whereas autoregressive training
results in matrices characterized by directionality and column dominance. The findings are then verified empirically, and inspired a symmetric initialization to speedup training of encoder-only models.

**Claims And Evidence:**

Yes.

**Essential References Not Discussed:**

n/a

**Experimental Designs Or Analyses:**

The experiments seem solid to me.

**Methods And Evaluation Criteria:**

Yes

**Other Comments Or Suggestions:**

n/a

**Other Strengths And Weaknesses:**

Strength: How training objective affects attention is an interesting yet underexplored question. The results in this paper are very interesting and provide both theory and practical values. I also like the intuition described in line 171-186.

Weakness: Given there is still about one-page space left in the draft, I think it would be better to give a formal theorem statement for Theorem 2.3 and 2.4 since they are the **main** results of the paper. The current way of presenting in my opinion, hurts the results's significance and rigor.

**Questions For Authors:**

1. Is the analysis in Section 2 particular to one-layer attention? Do you have any idea why certain layers of a model do not follow the findings of paper? (for decoder-only models, for example,  early layers are very symmetric). Would symmetric initialization of initial layers also speedup training of decoder-only models?

2. Could you specify the name of the model and the layer number in Figure 2?

**Relation To Broader Scientific Literature:**

Attention mechanism is a very sophisticated system with many moving parts, making it very opaque. The paper analyzes how training objective affects attention, which I think is a very important problem to study, and the results are very interesting.

**Theoretical Claims:**

i did not check the proofs closely.

---

> ### Author Rebuttal · Authors · 2025-04-01
>
> We thank the reviewer for the thorough and positive review, as well as the valuable suggestions and questions.
>
> First, we agree with the reviewer that we could have utilized the available manuscript space more effectively. In response to suggestions from Reviewers HwPP and JKhN, we will revise Section 2 by explicitly presenting Theorems 2.3 and 2.4, clearly stating their assumptions, conditions, and implications, integrating key points into the main text. We appreciate the reviewer’s feedback, as these adjustments will enhance both the rigor and clarity of our theoretical contributions.
>
>
> Second, we address the questions raised by the reviewer:
>
> 1. The analysis presented in Section 2 is not specific to single-layer self-attention, but applies generally to any layer within a Transformer model. Empirically, however, we observe that early layers can deviate from our theoretical predictions compared to deeper layers. This is expected, as backpropagation causes early layers to receive noisier updates, leading to greater divergence from theoretical predictions. On the other hand, the connections between tokens $x_i$ and $x_j$ and the corresponding prediction errors encoded by $\beta_{ij}$ are accurately reflected in the weight updates of deeper layers (which are closer to the output). The degree of deviation depends on the model and dataset and is often minimal. Figure S4 (complementing Figure 3) shows examples of models where such deviations are negligible. Based on these observations, we hypothesize that symmetric initialization of early layers is unlikely to be beneficial. Indeed, we tested symmetric initialization of all layers of decoder-only models, and we did not observe any significant improvement in training speed compared to standard initialization. We refer to our answer to question 1 of reviewer jZNG for further details.
>
> 2. We thank the reviewer for pointing this out. Since Figure 2 presents the symmetry and directionality scores for pretrained models, we assume this comment refers to Figure 3. We will update the caption of Figure 3 to explicitly mention the model (`Bert-Base-Uncased`) and the number of layers (12).

---

### Official Review · Reviewer_jZNG · 2025-03-11

**Overall Recommendation:** 4

**Summary:**

The paper investigates the inherent structures within self-attention mechanisms, focusing on symmetry, directionality, and emergent dynamics in Transformer training. The authors provide a mathematical framework for analyzing self-attention weight matrices and examine how different training objectives, namely bidirectional and autoregressive training, impact these structures. They argue that bidirectional training induces symmetric weight matrices, while autoregressive training results in directionality and column dominance. These findings are supported through theoretical derivations and empirical validation across multiple Transformer models.

**Claims And Evidence:**

The main claims of the paper include:

- Self-attention matrices exhibit structural properties influenced by the training objective.

- Bidirectional training promotes symmetry in weight matrices, whereas autoregressive training enforces directionality.

- These structural differences emerge naturally and can be leveraged to improve model performance.

The paper supports these claims through formal mathematical proofs and extensive experiments, showing consistent patterns across different model families and datasets.

**Essential References Not Discussed:**

The paper does not discuss some recent works on mechanistic interpretability of self-attention, such as studies on learned feature representations in attention layers. Including a discussion of these works could provide additional context and comparisons.

**Experimental Designs Or Analyses:**

The experimental design is sound, involving multiple Transformer architectures trained under different conditions. The authors analyze pretrained models to confirm the presence of symmetry and directionality trends. The statistical significance of results is demonstrated through interquartile range analyses. However, the study could be strengthened by including additional ablation studies to further isolate the effects of training objectives from other hyperparameters.

**Methods And Evaluation Criteria:**

The authors utilize a combination of theoretical analysis and empirical evaluation. They propose symmetry and directionality scores to quantify the structural properties of self-attention weight matrices. These metrics are applied to pretrained models, with comparisons across architectures trained on different datasets. The evaluation criteria are appropriate, as they align well with the research questions posed.

**Other Comments Or Suggestions:**

- Consider adding an appendix section with additional visualizations for symmetry and directionality scores across more layers.

- Clarify the potential implications of these findings for fine-tuning large-scale models.

**Other Strengths And Weaknesses:**

Strengths:

- The paper provides a novel theoretical perspective on self-attention structures.

- The empirical validation is thorough, covering multiple model families and training settings.

- The findings have potential applications in improving Transformer interpretability and efficiency.

Weaknesses:

- The paper does not explore practical implementations of the proposed insights, such as real-world deployment benefits.

- Some of the notation in the proofs is dense and may be challenging for readers unfamiliar with advanced linear algebra.

**Questions For Authors:**

1. Have you tested the impact of symmetric initialization on autoregressive models? If so, how does it compare to encoder-only models?

2. Could your framework be extended to analyze cross-attention mechanisms in encoder-decoder models?

3. Do your findings suggest any potential modifications to existing Transformer architectures for efficiency improvements?

**Relation To Broader Scientific Literature:**

This work builds on prior research in Transformer interpretability and self-attention mechanisms.

**Theoretical Claims:**

The paper presents rigorous mathematical derivations supporting its claims. Key results include:

- Proofs showing how gradient updates reinforce symmetric or directional structures depending on the training objective.

- Theorems explaining the emergence of column dominance in autoregressive models.

These theoretical insights are well-motivated and correctly formulated. The paper provides clear logical progressions from assumptions to conclusions, making the theoretical claims compelling.

---

> ### Author Rebuttal · Authors · 2025-04-01
>
> We thank the reviewer for the careful, detailed, and valuable input. First, we address the weaknesses raised by the reviewer:
>
> 1. We acknowledge that we did not evaluate real-world deployments, as our goal was to analyze the structures that emerge in self-attention matrices during pretraining. However, our findings suggest potential downstream benefits worth exploring in future work. Indeed, symmetric initialization speeds up training and leads to lower final loss, which often correlates with better downstream performance.
>
> 2. We thank the reviewer for highlighting this point, and we agree that the clarity of some notations could be improved. We will revise our proofs accordingly, specifically Proposition A.10.
>
> Next, we address the questions raised by the reviewer:
>
> 1. We have conducted symmetric initialization experiments on autoregressive models. Our observations indicate that these models quickly lose their initial symmetry, eventually converging to $W_{qk}$ matrices with low symmetry scores. Furthermore, we observed no significant improvement in training speed compared to standard initialization. For details, please refer to the plots of the loss curve and symmetry available here (12 layers model): https://drive.google.com/file/d/14Y8huSc7EajiLWiGjjQNM3-BPvRVCw-G/view?usp=sharing
>  In contrast, as shown in the manuscript, encoder-only models clearly benefited from symmetric initialization, exhibiting faster training and higher overall symmetry scores.
>
> 2. We thank the reviewer for raising this interesting point. We have not yet explored how to extend our mathematical framework to cross-attention mechanisms. Nonetheless, our empirical results show that the encoder components of encoder-decoder models consistently exhibit higher symmetry scores than the decoder components, and this difference holds across models of varying sizes (see Figure S2). This empirical evidence suggests that Theorem 2.4 can potentially be extended to cross-attention.
>
> 3. Our results highlight that the implicit weight updates of the $W_{qk}$ matrix encode essential structures for Transformer training. We hypothesize that architectural designs that better preserve and reinforce these structures in both the column and row spaces of $W_{qk}$ could further enhance training efficiency.
>
> On the other comments and suggestions:
>
> 1. We respectfully ask the reviewer for clarification regarding which additional layers should be visualized. Figures 3 and S4 already show symmetry and directionality scores across all layers of several models trained from scratch on three distinct datasets.
>
> 2. The main goal of our study was to characterize the structures that emerge in self-attention matrices during pretraining. As discussed above, future work should explore how symmetric initialization can be leveraged to potentially benefit from fine-tuning on downstream tasks.

---

### Official Review · Reviewer_JKhN · 2025-03-14

**Overall Recommendation:** 3

**Summary:**

This work investigates the training process of Transformers, revealing a structured pattern of the attention weights' update as a linear combination of rank-1 matrices. Based on this, it is demonstrated that bidirectional training (encoder-only) induces symmetry in weight matrices, while autoregressive training (decoder-only) results in directionality and column dominance. These phenomena are both mathematically proved and numerically verified.

## Update after rebuttal

Authors have updated further experiments on large-scale language modeling tasks to demonstrate the effectiveness of symmetry initializations. However, this technique is probably not that valid on image datasets, which limits the applicability of the derived (core) insights of this work. Given the theoretical and (partial) practical contributions, I maintained the recommendation to be not against acceptance.

**Claims And Evidence:**

The claims made in the submission are supported by clear and convincing evidence.

**Essential References Not Discussed:**

To the best of my knowledge, I am not aware of any essential references not discussed in this submission.

**Experimental Designs Or Analyses:**

The experimental designs/analyses in this submission align with and validate corresponding theoretical claims.

**Methods And Evaluation Criteria:**

The proposed initialization strategy is inspired by the weights' symmetry pattern for bidirectional training, and hence quite reasonable.

**Other Comments Or Suggestions:**

Minor issues:

1. Eq. (12): $[\textbf{x}_i]_k$ -> $[\textbf{x}_i]_m$.
2. Section 6: $\textbf{W}qk$ -> $\textbf{W}_{qk}$.
3. Line 1286: "smaller" -> "larger".

**Other Strengths And Weaknesses:**

Strengths:

1. This paper is well-written and easy to follow.
2. There are rigorous mathematical analysis and consistent numerical verifications.

Weaknesses:

1. The paper length is around 7 pages, allowing to present more details in the main texts.
2. Some of the current statements/discussions are repetitive and can be reduced (see details in the following "Questions For Authors" section).

**Questions For Authors:**

1. Proposition 2.1: It seems that Eq. (3) is just an enrollment of residual (single-head) self-attention, and Eq. (5) trivially holds due to the monotonicity of softmax operations. What is the significance of this proposition?
2. Table 1: Note that the performance enhancements of symmetry initializations significantly degrade as the model depth increases. It would be more convincing to test for larger models. In addition, current experiments are all conducted on language datasets. Do similar enhancements appear for image datasets?
3. Besides symmetry initialization, is it possible to also explore (possibly layer-wise) symmetry regularizations for further enhancements?
4. How do we exploit the directionality pattern to improve the autoregressive training?

**Relation To Broader Scientific Literature:**

This work generalizes prior results (e.g. Trockman & Kolter (2023)) to initialize query-key matrices as identities based on uncovered diagonal  patterns.

**Theoretical Claims:**

I did not check every detail of mathematical proofs, but the theoretical derivations seem sound.

---

> ### Author Rebuttal · Authors · 2025-04-01
>
> We thank the reviewer for the detailed and constructive feedback provided.
>
> Below, we address the reviewer’s questions, along with the related concerns highlighted as potential weaknesses.
>
> 1. Proposition 2.1 aimed to show that accurate token prediction depends on learning an effective bilinear form in the embedding space, represented by the implicit matrix product $\mathbf{W}_{qk}$. However, we agree with the reviewer that this proposition is not essential. We will revise Section 2 accordingly by: (a) removing Proposition 2.1 and integrating its key insights into the main text, and (b) expanding Section 2.3 and adding a new Section 2.4 to present Theorems 2.3 and 2.4 in greater detail, making better use of the available space.
>
> 2. We thank the reviewer for raising these points. The comments led to valuable additional experiments and analyses that helped strengthen the empirical support for our findings. To provide a clear response, we address the two questions separately:
>
> - 2.1 We conducted additional experiments using larger models and observed a consistent training speed-up, in line with our previous results. Specifically, we trained a BERT-large model (24 layers, 3x more parameters than Bert-Base) on the same three datasets used for BERT-mini and BERT-base, that is, Jigsaw, Wikipedia, and Red Pajama. Although full training runs are set to 200k steps, the current results are based on intermediate checkpoints. For Wikipedia, after 123k steps, the loss decreased from 0.2151 (without symmetric initialization) to 0.1874 (with symmetric initialization), yielding a 56.5% speed-up. For Jigsaw, after 184k steps, the loss dropped from 0.8113 to 0.7612, with an 11.0% speed-up. Similarly, for Red Pajama, after 140k steps, the loss improved from 0.2261 to 0.2058, with a 37.8% speed-up. We will include the final results in Table 1 of the revised manuscript. Furthermore, we found a clear positive correlation between increased symmetry from symmetric initialization and both faster training and lower final loss. For details, please refer to: https://drive.google.com/file/d/1nuAGfyjVAp9suVL0NydVPOTQK56_AtvP/view?usp=sharing. Finally, while the reviewer correctly notes a smaller relative gain in speed-up from BERT-mini to BERT-base, this is expected due to neural scaling laws (e.g., Kaplan et al., 2020) that make improvements harder at larger scales.
>
> - 2.2 We conducted preliminary experiments on training Vision Transformers on the CIFAR-10 and ImageNet-1k datasets. We did not observe a significant speed-up with these specific experiments. Nonetheless, we hypothesize that further analysis is necessary to check if symmetric initialization can speed up the training of vision Transformers.
>
> 3. We have conducted experiments to enforce symmetry across layers by adding a regularization loss term, as follows:
>
> $L_{reg} = \frac{\| \mathbf{W}^l_{qk}\|}{\| {\mathbf{W}^l_{qk}}_s\|} \quad \forall l \in [0,L] ,$
>
> where the denominator is the symmetric component of the $\mathbf{W}^l_{qk}$ matrix. However, this did not lead to noticeable improvements in training speed or final loss compared to the baseline. While we see training constraints that promote symmetry as a promising direction for future work, the specific regularization method we tested was not effective. For details on our results, please refer to: https://drive.google.com/file/d/1Paa7z6KxD11MdXyJyhU-xOUFqg6tJkSf/view?usp=share_link
>
> 4. We thank the reviewer for raising this important point. In our current work, we have successfully explored methods to exploit symmetry. We are investigating several approaches to leverage column dominance to improve autoregressive training. These ongoing experiments will be fully addressed and presented in a dedicated future study.
>
> On the other comments and suggestions, we thank the reviewer for pointing out 3 typos in the manuscript. We will revise the manuscript accordingly.

---

### Official Review · Reviewer_HwPP · 2025-03-19

**Overall Recommendation:** 2

**Summary:**

In this paper the authors focus on the structure of the attention matrix used in Transformers and in particular the effect of the training strategy on the overall structure inherited by the same. Showcasing that autoregressive training leads to directional matrices, whereas  bidirectional training induces symmetry, these insights are tested on a wide array of practical models.

**Claims And Evidence:**

Yes

**Essential References Not Discussed:**

No

**Experimental Designs Or Analyses:**

Yes

**Methods And Evaluation Criteria:**

Yes

**Other Comments Or Suggestions:**

See above

**Other Strengths And Weaknesses:**

Strengths:

1. I think the authors study an important problem with regards to understanding attention structures and the final takeaways regarding their symmetry are quite interesting. In particular, trying symmetric initialization strategies and showing speedup is quite nice.

2. I also appreciate the mathematical effort to compute the gradients and reorient them in a suitable way to obtain important insights about the directionality and symmetry of the attention matrices.

Weaknesses:

1. Mathematical rigor part highlighted above. Also the text explanation of the mathematical results is not fully satisfactory. It requires lot more polishing in translating the importance of intuition behind these results into a coherent set of paragraphs.
2. Likewise, I feel the paper could benefit a lot from significant rewriting by clearly stating what's the setup for theoretical results is. And how and why these insights translate to practical experiments. Currently, it's unclear for what input data or scenarios, the results Theorems 2.3 and 2.4 hold and why they should translate to real data. Improving upon these points can help make the paper more crisp and direct.

**Questions For Authors:**

1. If I understand correctly, in Section 4.2 either you initialize randomly or with symmetry and let them free to train right? Is there any reason why you didn't try keeping it symmetric throughout the training? What happens in this scenario? For example, you can initialize $W_k = W_q$.

2. Is symmetry score =1 the ideal scenario for bidirectional training? Or is there a reason why this is not always the best?

**Relation To Broader Scientific Literature:**

Authors cited all the relevant literature in connection to the paper

**Theoretical Claims:**

I felt that the paper lacked the prerequisite mathematical rigor for the claims and propositions in the main paper. In fact I found an important mistake about bi-directional training loss which I am not sure how it will change the main results. That is, the Equation (S8) in the appendix is not true. Take $N=2$ for example. The factorization of $P(t_1, t_2) = P(t_1|t_2) \cdot P(t_2|t_1)$ is simply not true unless $t_1$ and $t_2$ are independent, which is not stated.

With regards to lack of mathematical rigor, following are some examples:

1. What's the assumption on the input data for main theorems 2.3 and 2.4 to hold? Does it hold for all inputs? What's the probability over in Equation (13)?
2. Likewise what's $P_j$  in Proposition 2.2. I only got to know its meaning in the appendix.
3. Proposition 2.1 is a simple consequence of the fact that the linear attention scores and soft-max attention scores follow the same ordering because the latter is just the exponential of the former up to some scaling. So in the current form it sounded like a bit of fancy result with jargon like projections, subspaces and I didn't really see the need for it nor the main importance of this result, despite the explanation below which didn't fully sound satisfactory to me.

---

> ### Author Rebuttal · Authors · 2025-04-01
>
> We thank the reviewer for the careful review, for going through the proofs in the Appendix, and for the detailed feedback to improve clarity. While we appreciate the concerns raised, we respectfully disagree with the claim that the manuscript lacks the necessary mathematical rigor for its claims and propositions, and we outline our reasoning below.
>
> First, we acknowledge the mistake in Equation (S8), but this has no impact on the correctness of our proofs or results. Indeed, like with Equation (S6), (S8) was intended as a direct factorization of the joint distribution under bidirectional training. However, it was included only to motivate Equations (S9) and (S10) (the standard Masked Language Modeling (MLM) objective) which is what we use throughout the paper, and does not depend on (S8). We will remove (S8) and keep only (S9) and (S10).
>
> Second, we address the weaknesses regarding the explanation of our mathematical results (1-2). We value the reviewer’s feedback and have restructured the presentation of our results to improve clarity. We emphasize that our core results focus on deriving the implicit gradient of self-attention matrices, under minimal assumptions about the data. Importantly, the results in Sections 2.1–2.2 and Proposition A.7 rely solely on properties of self-attention. We are confident that the following revisions will better highlight the intuitions and clarify the theoretical setup:
>
> - Section 2.1: We will keep the standard definition of self-attention and keep Equation (3) to connect it later with (8), (9), and Figure 1. Proposition 2.1 will be removed entirely, and only the essential content needed to link (3) to Proposition 2.2 will be preserved. Given these structural changes, we agree with the reviewer that Proposition 2.1 is not necessary for understanding the subsequent results. We will retain the reference to Section A.1 for relevant definitions and remove the proof of Proposition 2.1.
>
> - Section 2.2: We will present a general definition of the negative log-likelihood in Equation (6) without introducing $C_i$ at that point. Then, in Proposition 2.2, we will define both $C_i$ and $P_j$, followed by a clear explanation of how the gradient of $W_{qk}$ can be derived using these two equivalent summations. This will provide the foundation for introducing the two main theorems.
>
> - Section 2.3: We will enhance the explanation of how a token contributes to the gradient when used as context or prediction by explicitly including Proposition A.7 in this section. Up to this point, all mathematical results are derived purely from self-attention and do not assume anything about the input data. We will move the theorems to a new Section 2.4.
>
> - Section 2.4: We will formally state Theorems 2.3 and 2.4 with their assumptions: (1) There are statistical correlations between tokens, a weak and general assumption that holds in most real-world data. As a result, token embeddings exhibit partial alignment, capturing semantic and predictive structure. Indeed, this alignment either exists in pretrained embeddings or naturally emerges during training in learned embeddings, encoding semantic relationships; (2) Entries of $W_{qk}$ are i.i.d. at initialization with finite mean and variance; (3) Bidirectional training induces approximately symmetric error signals, that is, the error in predicting token $i$ from $j$ is similar to that of predicting $j$ from $i$. This new section will clarify why the theoretical setup broadly applies to real-world data and what predictions it enables.
>
> - Section 3-4: The current versions demonstrate how the predicted structures appear in Transformer models (Fig. 2 for language; Fig. S1 for vision and audio), emerge during training (Fig. 3), and can be leveraged in practical applications (Table 1). These results clearly show how our theoretical insights translate to real-world scenarios.
>
> Finally, we address the questions raised by the reviewer:
> 1.  Yes, during symmetric initialization, we randomly initialize $W_q$ and set $W_k = W_q$, ensuring $W_qW_k^T$ is symmetric, as we understand the reviewer suggests. We also experimented with a regularization term to enforce symmetry during training, but it did not improve training speed or final loss compared to the baseline. For details, see point 3 in our response to reviewer jKhN.
>
> 2. Our framework shows that, under bidirectional training, each token pair contributes to an approximately symmetric update of $W_{qk}$. However, it does not determine whether a fully symmetric $W_{qk}$ is "ideal," as this would require a precise definition of "ideal" and an analysis of attention matrices at convergence. This is an interesting direction for future work. Additionally, Remark A.15 highlights that, in MLM, only a subset of updates is symmetric. As a result, we naturally expect, and have empirically observed, non-fully symmetric $W_{qk}$, though these still exhibit significantly higher symmetry scores than those from autoregressive training.

---

### Decision · Program_Chairs · 2025-05-01

**Decision:**

Accept (poster)

**Comment:**

The paper studies how bidirectional and autoregressive training objectives influence the structure of the query-key matrix in self-attention. The results show that bidirectional training induces symmetric structures, whereas autoregressive training results in matrices characterized by directionality and column dominance. The findings are then verified empirically, and inspired a symmetric initialization to speedup training of encoder-only models.

With scores 2,3,4,4, and after carefully reading the comments and authors' rebuttal, I recommend acceptance.

There is strong support from three reviewers. The reviewer recommending rejection had raised concerns about incorrectness of one equation and more importantly the lack of mathematical rigor as manifested in the absence of explicit statement of assumptions and full version of theorems deferred to appendix or very simple results stated as propositions.
The authors have:
(i) clarified that the incorrect equation is not affecting any of the conclusions and was in a non-critical equation meant to give intuition which will be removed
(ii) responded with a clear plan to restructure Section 2 and formalize statements

Since other reviewers who eventually recommend acceptance also discussed the issues about formalizing statements and bringing all details from appendix to main body and appear content with the authors' promise to do that, I recommend acceptance.

However, I do urge the authors to include all the required modifications in the camera ready. These include:
(i) Give formal theorem statements for Theorem 2.3 and 2.4 in main body (ii) Make other promised changes to structure of Section 2 (including clarifying Prop. 2.1, expanding Section 2.3) (iii) Add additional experiments on large-scale datasets as stated in rebuttal (including those on vision models despite not revealing gains)